# PARALLELIZING NON-LINEAR SEQUENTIAL MODELS OVER THE SEQUENCE LENGTH

**Y. H. Lim**[1], **Q. Zhu** [1], **J. Selfridge**[2][*] **M. F. Kasim**[1][†]
[1] Machine Discovery Ltd., UK, [2] University of Oxford, UK
{yi.heng, qi.zhu, muhammad}@machine-discovery.com
joshua.selfridge@trinity.ox.ac.uk

## ABSTRACT

Sequential models, such as Recurrent Neural Networks and Neural Ordinary Differential Equations, have long suffered from slow training due to their inherent sequential nature. For many years this bottleneck has persisted, as many thought sequential models could not be parallelized. We challenge this long-held belief with our parallel algorithm that accelerates GPU evaluation of sequential models by up to 3 orders of magnitude faster without compromising output accuracy. The algorithm does not need any special structure in the sequential models' architecture, making it applicable to a wide range of architectures. Using our method, training sequential models can be more than 10 times faster than the common sequential method without any meaningful difference in the training results. Leveraging this accelerated training, we discovered the efficacy of the Gated Recurrent Unit in a long time series classification problem with 17k time samples. By overcoming the training bottleneck, our work serves as the first step to unlock the potential of non-linear sequential models for long sequence problems.

## 1 INTRODUCTION

Parallelization is arguably a main workhorse in driving the rapid progress in deep learning over the past decade. Through specialized hardware accelerators such as GPU and TPU, matrix multiplications which are prevalent in deep learning can be evaluated swiftly, enabling rapid trial-and-error in research. Despite the widespread use of parallelization in deep learning, sequential models such as Recurrent Neural Networks (RNN) (Hochreiter & Schmidhuber, 1997; Cho et al., 2014) and Neural Ordinary Differential Equations (NeuralODE) (Chen et al., 2018; Kidger et al., 2020) have not fully benefited from it due to their inherent need for serial evaluations over sequence lengths.

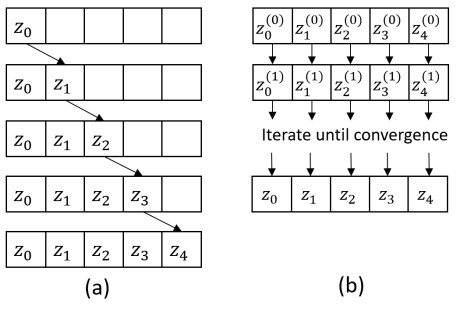

Figure 1: Evaluating sequential models using (a) sequential method and (b) iterative method that is parallelizable.

Serial evaluations have become the bottleneck in training sequential deep learning models. This bottleneck might have diverted research away from sequential models. For example, attention mechanism (Bahdanau et al., 2014) and transformers (Vaswani et al., 2017) have dominated language modelling over RNN in recent years, partly due to their ability to be trained in parallel (Hooker, 2021). Continuous Normalizing Flows (CNF) (Chen et al., 2018; Grathwohl et al., 2018), which used to utilize NeuralODE as their models, has been moving towards the direction where the training does not involve simulating the ODE (Lipman et al., 2022; Rozen et al., 2021; Ben-Hamu et al., 2022). More recent works (Orvieto et al., 2023;

---

[*]Work done during internship at Machine Discovery
[†]Corresponding author

Huang et al., 2022) have attempted to resurrect the sequential RNN, but they focus on linear recurrent layers that can be evaluated in parallel with prefix scan (Blelloch, 1990; Martin & Cundy, 2018; Smith et al., 2022), leaving non-linear recurrent layers unparallelizable over their sequence length.

In this paper, we present an algorithm that can parallelize the evaluation and training of non-linear sequential models like RNN and NeuralODE without changing the output of the models beyond reasonable numerical precision. We do this by introducing a general framework to solve non-linear differential equations by restating them as fixed-point iteration problems with quadratic convergence, equivalent to Newton's method for root finding. The fixed-point iteration involves parallelizable operations and an inverse linear operator that can be evaluated in parallel even for sequential models like RNN and ODE (see Figure 1). As the convergence is quadratic, the number of fixed-point iterations can be quite small especially when the initial starting point is close to the converged solution. Most importantly, the proposed algorithm does not need a special structure of sequential models, removing the need to change models' architecture to be able to take advantage of the parallelization.

## 2 RELATED WORKS

There have been some attempts to parallelize the evaluation and training of sequential models, especially for RNN. However, most (if not all) of them require special structures of recurrent layers. Lei et al. (2017) changed the matrix multiplication involving the states into element-wise multiplication to enable parallelization. Luo et al. (2020) segments the sequence into several groups to be evaluated by RNN in parallel and their groups interdependency are learned by a higher-level RNN. Huang et al. (2022), Orvieto et al. (2023), and Martin & Cundy (2018) use linear recurrent layer where it can be evaluated in parallel using prefix scan (Blelloch, 1990).

On the NeuralODE side, the parallelization effort mainly comes from the past works in parallelizing the ODE solver. One of the main idea is the multiple shooting method (Kiehl, 1994; Gander & Vandewalle, 2007; Chartier & Philippe, 1993; Bellen & Zennaro, 1989; Lions et al., 2001) where the time sequence is split into several segments then multiple ODE solvers are executed for each segment in parallel. The process is repeated iteratively until the solutions from all segments are matched. Although multiple ODE solvers can be executed in parallel, each ODE solver itself still requires sequential operation. The multiple shooting method has been adapted in training NeuralODE in Massaroli et al. (2021). The multi-grid idea is also applied to parallelize a special RNN unit and ResNet by converting them to an ODE (Gunther et al., 2020; Moon & Cyr, 2022). Another work, neural rough ODE (Morrill et al., 2021) has shown that it is possible to train NeuralODE for very long sequence by computing log-signature of the signals over large steps and enable the ODE solver to take a big step. The calculation of the log-signature can be done in parallel. However, solving the ODE still requires sequential operation even though the time step is larger than the original.

Over the past few years, there have been increasing interests in using fixed points finders (e.g., root-finder) in deep learning. DEQ (Bai et al., 2019; 2021) employs root-finding in evaluating neural networks with infinite identical layers. The concept of infinite layers paired with root-finders has been applied to various problems and has yielded impressive results (Bai et al., 2020; Liu et al., 2022; Huang et al., 2021). Nonetheless, these works do not talk about parallelizing sequential models.

Several recent works have also looked into parallelizing sequential models using fixed-point iterations or root finders. In the context of stochastic generative modelling, Shih et al. (2023) divides the time span into several regions and then employs Picard iteration that are parallelizable in solving the ODE. Wang & Ragni (2021) recast RNN evaluation as a fixed-point iteration and solve it by only doing a small number of iterations without checking the convergence. As a result, this approach might produce different results than evaluating RNN sequentially. Song et al. (2021) evaluates feedforward computations by solving non-linear equations using Jacobian or Gauss-Seidel iterations. Notably, the works mentioned above only use zeroth order fixed-point iterations that converge slower than our method and might not be able to converge at all if the mapping is not contracting.

## 3 DEER FRAMEWORK

We will present the DEER framework: "non-linear Differential Equation as fixed point itERation" with quadratic convergence and show its relation to Newton's method. This framework can be ap-

plied to 1D differential equations, i.e., ODEs, as well as differential equations in higher dimensions, i.e. Partial Differential Equations (PDEs). The same framework can also be adopted to discrete difference equations to achieve the same convergence rate, which can be applied to RNN. With the framework, we can devise a parallel algorithm to evaluate RNN and ODE without significant changes to the results.

## 3.1  DEER FRAMEWORK

Consider an output signal of interest $\mathbf{y}(\mathbf{r}) \in \mathbb{R}^n$ which consists of $n$ signals on a $d$-dimensional space, where the coordinate is denoted as $\mathbf{r} \in \mathbb{R}^d$. The output signal, $\mathbf{y}(\mathbf{r})$, might depend on an input signal, $\mathbf{x}(\mathbf{r})$, via some non-linear delayed differential equation (DE),

$$L[\mathbf{y}(\mathbf{r})] = \mathbf{f}\left(\mathbf{y}(\mathbf{r} - \mathbf{s}_1), \mathbf{y}(\mathbf{r} - \mathbf{s}_2), ..., \mathbf{y}(\mathbf{r} - \mathbf{s}_P), \mathbf{x}(\mathbf{r}), \theta\right) \tag{1}$$

where $L[\cdot] : \mathbb{R}^n \to \mathbb{R}^n$ is the linear operator of the DE, and $\mathbf{f}$ is the non-linear function that depends on values of $\mathbf{y}$ at $P$ different locations, external inputs $\mathbf{x}$, and parameters $\theta$. This form is general enough to capture various continuous differential equations such as ODE (with $L[\cdot] = d/dt$ and $\mathbf{r} = t$), partial differential equations (PDEs), or even a discrete difference equations for RNN.

Now let's add $\mathbf{G}_p(\mathbf{r})\mathbf{y}(\mathbf{r} - \mathbf{s}_p)$ terms on the left and right hand side, where $\mathbf{G}_p(\mathbf{r}) : \mathbb{R}^{n \times n}$ is an $n$-by-$n$ matrix that depends on the location $\mathbf{r}$. The values of $\mathbf{G}_p$ will be determined later. Equation 1 now becomes

$$L[\mathbf{y}(\mathbf{r})] + \sum_{p=1}^{P} \mathbf{G}_p(\mathbf{r})\mathbf{y}(\mathbf{r} - \mathbf{s}_p) = \mathbf{f}\left(\mathbf{y}(\mathbf{r} - \mathbf{s}_1), ..., \mathbf{x}(\mathbf{r}), \theta\right) + \sum_{p=1}^{P} \mathbf{G}_p(\mathbf{r})\mathbf{y}(\mathbf{r} - \mathbf{s}_p) \tag{2}$$

$$\mathbf{y}(\mathbf{r}) = L_{\mathbf{G}}^{-1}\left[\mathbf{f}\left(\mathbf{y}(\mathbf{r} - \mathbf{s}_1), ..., \mathbf{x}(\mathbf{r}), \theta\right) + \sum_{p=1}^{P} \mathbf{G}_p(\mathbf{r})\mathbf{y}(\mathbf{r} - \mathbf{s}_p)\right] \tag{3}$$

The left hand side of equation 2 is a linear equation with respect to $\mathbf{y}$, which can be solved more easily than solving the non-linear equation in most cases. In equation 3, we introduce the notation $L_{\mathbf{G}}^{-1}[\cdot]$ as a linear operator that solves the linear operator on the left-hand side of equation 2 with some given boundary conditions.

Equation 3 can be seen as a fixed-point iteration problem, i.e., given an initial guess $\mathbf{y}^{(0)}(\mathbf{r})$, we can iteratively compute the right hand side of the equation until it converges. To analyze the convergence near the true solution, let's denote the value of $\mathbf{y}$ at $i$-th iteration as $\mathbf{y}^{(i)}(\mathbf{r}) = \mathbf{y}^*(\mathbf{r}) + \delta\mathbf{y}^{(i)}(\mathbf{r})$ with $\mathbf{y}^*(\mathbf{r})$ as the true solution that satisfies equation 3. Putting $\mathbf{y}^{(i)}$ into equation 3 to get $\mathbf{y}^{(i+1)}$, and performing Taylor expansion up to the first order, we obtain

$$\delta\mathbf{y}^{(i+1)}(\mathbf{r}) = L_{\mathbf{G}}^{-1}\left[\sum_{p=1}^{P} [\partial_p\mathbf{f} + \mathbf{G}_p(\mathbf{r})]\,\delta\mathbf{y}^{(i)}(\mathbf{r} - \mathbf{s}_p) + O(\delta\mathbf{y}^2)\right] \tag{4}$$

where $\partial_p\mathbf{f}$ is the Jacobian matrix of $\mathbf{f}$ with respect to its $p$-th parameters. From the equation above, the first order term of $\delta\mathbf{y}^{(i+1)}$ can be made 0 by choosing

$$\mathbf{G}_p(\mathbf{r}) = -\partial_p\mathbf{f}\left(\mathbf{y}(\mathbf{r} - \mathbf{s}_1), ..., \mathbf{y}(\mathbf{r} - \mathbf{s}_P), \mathbf{x}(\mathbf{r}), \theta\right). \tag{5}$$

It shows that the fastest convergence around the solution can be achieved by choosing the matrix $\mathbf{G}_p$ according to the equation above. It can also be shown that the iteration in equation 3 and equation 5 is equivalent to the realization of Newton's method in Banach space, therefore offering a quadratic convergence. The details of the relationship can be found in Appendix A.1 and the proof of quadratic convergence can be seen in Appendix A.3.

Iterative process in equation 3 involves evaluating the function $\mathbf{f}$, its Jacobian, and matrix multiplications that can be parallelized in modern accelerators (such as GPU and TPU). If solving the linear equation can be done in parallel, then the whole iteration process can take advantage of parallel computing. Another advantage of solving non-linear differential equations as a fixed point iteration problem in the deep learning context is that the solution from the previous training step can be used as the initial guess for the next training step if it fits in the memory. Better initial guess can lower the number of iterations required to find the solution of the non-linear DE.

### 3.1.1 DERIVATIVES

To utilize the framework above in deep learning context, we need to know how to calculate the forward and backward derivatives. For the forward derivative, we would like to know how much $\mathbf{y}$ is perturbed (denoted as $\delta\mathbf{y}$) if the parameter $\theta$ is slightly perturbed by $\delta\theta$. By applying the Taylor series expansion to the first order to equation 1 and using $\mathbf{G}_p(\mathbf{r})$ as in equation 5, we obtain

$$\delta\mathbf{y} = L_{\mathbf{G}}^{-1}\left[\frac{\partial\mathbf{f}}{\partial\theta}(\mathbf{y}(\mathbf{r} - \mathbf{s}_1), ..., \mathbf{y}(\mathbf{r} - \mathbf{s}_P), \mathbf{x}(\mathbf{r}), \theta)\delta\theta\right]. \tag{6}$$

The equation above means that to compute $\delta\mathbf{y}$, one can compute the forward derivative of the outputs of $\mathbf{f}$ from $\delta\theta$ (denoted as $\delta\mathbf{f}$), then execute the inverse linear operator $L_{\mathbf{G}}^{-1}$ on $\delta\mathbf{f}$.

For the backward derivative, the objective is to obtain the gradient of a loss function $\mathcal{L}$ with respect to the parameter, $\partial\mathcal{L}/\partial\theta$, given the gradient of the loss function to the output signal, $\partial\mathcal{L}/\partial\mathbf{y}$. To obtain the backward derivative, we write

$$\frac{\partial\mathcal{L}}{\partial\theta} = \frac{\partial\mathcal{L}}{\partial\mathbf{y}}\frac{\partial\mathbf{y}}{\partial\theta} = \left(\left(\frac{\partial\mathcal{L}}{\partial\mathbf{y}}L_{\mathbf{G}}^{-1}\right)\frac{\partial\mathbf{f}}{\partial\theta}(\mathbf{y}(\mathbf{r} - \mathbf{s}_1), ..., \mathbf{y}(\mathbf{r} - \mathbf{s}_P), \mathbf{x}(\mathbf{r}), \theta)\right). \tag{7}$$

Note that the order of evaluation in computing the backward gradient should follow the brackets in the equation above. In contrast to equations 3 and 6, the linear operator $L_{\mathbf{G}}^{-1}$ is operated to the left in the innermost bracket in equation 7. This is known as the dual operator of $L_{\mathbf{G}}^{-1}$ and in practice can be evaluated by applying the vector-Jacobian product of the linear operator $L_{\mathbf{G}}^{-1}[\cdot]$. The results of the dual operator in the innermost bracket are then the followed by vector-Jacobian product of the function $\mathbf{f}$. The forward and backward derivatives for $\mathbf{x}$ are similar to the expressions for $\theta$: just substitute the differential with respect to $\theta$ into the differential with respect to $\mathbf{x}$.

The forward and backward gradient computations involve only one operation of $L_{\mathbf{G}}^{-1}$, in contrast to the forward evaluation that requires multiple iterations of $L_{\mathbf{G}}^{-1}$ evaluations. This allows the gradient computations to be evaluated more quickly than the forward evaluation. Moreover, the trade-off between memory and speed can be made in the gradient computations. If one wants to gain the speed, the matrix $\mathbf{G}$ from the forward evaluation can be saved to be used in the gradient computation. Otherwise, the matrix $\mathbf{G}$ can be recomputed in the gradient computation to save memory.

The gradient equations above apply even if the forward evaluation does not follow the algorithm in the previous subsection. For example, if equation 3 does not converge and forward evaluation is done differently (e.g., sequentially for RNN), the backward gradient can still be computed in parallel according to equation 7. This might still provide some acceleration during the training.

### 3.2 PRACTICAL IMPLEMENTATION

Equation 1 has a very general form that can capture ordinary differential equations (ODEs), most partial differential equations (PDEs), and even discrete difference equations. To apply DEER framework in equation 3 to a problem, there are several steps that need to be followed. The first step is to recast the problem into equation 1 to define the variable $\mathbf{y}$, the linear operator $L[\cdot]$, and the non-linear function $\mathbf{f}(\cdot)$. The second step is to implement what we call the shifter function. The shifter function takes the whole discretized values of $\mathbf{y}(\mathbf{r})$ and returns a list of the values of $\mathbf{y}$ at the shifted positions, i.e. $\mathbf{y}(\mathbf{r} - \mathbf{s}_p)$ for $p = \{1, ..., P\}$. The shifter function might need some additional information such as the initial or boundary conditions. The output of the shifter function will be the input to the non-linear function. The next step, and usually the hardest step, is to implement the inverse operator $L_{\mathbf{G}}^{-1}[\mathbf{h}]$ given the list of matrices $\mathbf{G}_p$ and the vector values $\mathbf{h}$ discretized at some points. The inverse operator $L_{\mathbf{G}}^{-1}[\mathbf{h}]$ might also need the information on the boundary conditions.

Once everything is defined, iterating equation 3 can be implemented following the code in appendix B.1. The Jacobian matrix in equation 5 can be calculated using automatic differentiation packages (Paszke et al., 2017; Frostig et al., 2018).

DEER framework can be applied to any differential or difference equations as long as the requirements above can be provided. This includes ordinary differential equations, discrete sequential models, and even partial differential equations (see Appendix A.4). To keep focused, we only present the application of DEER in parallelizing ODE and discrete sequential models.

### 3.3 Parallelizing ordinary differential equations (ODE)

An ODE typically takes the form of $d\mathbf{y}/dt = \mathbf{f}(\mathbf{y}(t), \mathbf{x}(t), \theta)$ where the initial condition $\mathbf{y}(0)$ is given. The ODE form above can be represented by equation 1 with $\mathbf{r} = t$, $L = d/dt$, $P = 1$, and $\mathbf{s}_1 = 0$. This means that the operator $L_{\mathbf{G}}^{-1}$ in ODE is equivalent to solving the linear equation below given the initial condition $\mathbf{y}(0)$,

$$\frac{d\mathbf{y}}{dt}(t) + \mathbf{G}(t)\mathbf{y}(t) = \mathbf{z}(t) \iff \mathbf{y}(t) = L_{\mathbf{G}}^{-1}[\mathbf{z}(t)]. \tag{8}$$

Assuming that $\mathbf{G}(t)$ and $\mathbf{z}(t)$ are constants between $t = t_i$ and $t = t_{i+1}$ as $\mathbf{G}_i$ and $\mathbf{z}_i$ respectively, we can write the relations between $\mathbf{y}_{i+1} = \mathbf{y}(t_{i+1})$ and $\mathbf{y}_i = \mathbf{y}(t_i)$ as

$$\mathbf{y}_{i+1} = \bar{\mathbf{G}}_i \mathbf{y}_i + \bar{\mathbf{z}}_i \tag{9}$$

with $\bar{\mathbf{G}}_i = \exp(-\mathbf{G}_i \Delta_i)$, $\bar{\mathbf{z}}_i = \mathbf{G}_i^{-1}(\mathbf{I} - \bar{\mathbf{G}}_i)\mathbf{z}_i$, $\Delta_i = t_{i+1} - t_i$, $\mathbf{I}$ the identity matrix, and $\exp(\cdot)$ the matrix exponential. Equation 9 can be evaluated using the parallel prefix scan algorithm as described in Blelloch (1990) and Smith et al. (2022). Specifically, we define a pair of variables $c_{i+1} = (\bar{\mathbf{G}}_i | \bar{\mathbf{z}}_i)$ for every discrete time point $t_i$, the initial values $c_0 = (\mathbf{I} | \mathbf{y}_0)$, and an associative operator,

$$c_{i+1} \bullet c_{j+1} = (\bar{\mathbf{G}}_j \bar{\mathbf{G}}_i | \bar{\mathbf{G}}_j \bar{\mathbf{z}}_i + \bar{\mathbf{z}}_j). \tag{10}$$

Given the initial value of $c_0$ and the associative operator above, we can run the associative scan in parallel to get the cumulative value of the operator above. The solution $\mathbf{y}_i$ can be taken from the second element of the results of the parallel scan operator.

On the implementation note, we can reduce the computational error by taking the $\mathbf{G}_i$ and $\mathbf{z}_i$ as the mid-point value, i.e. $\mathbf{G}_i = \frac{1}{2}[\mathbf{G}(t_i) + \mathbf{G}(t_{i+1})]$ and $\mathbf{z}_i = \frac{1}{2}[\mathbf{z}(t_i) + \mathbf{z}(t_{i+1})]$. By taking the mid-point value, we can obtain the third order local truncation error $O(\Delta_i^3)$ instead of second order error using either the left or the right value, with only a small amount of additional computational expenses. The details can be seen in Appendix A.5.

The materialization of DEER framework on ODE can be seen as direct multiple shooting method (Chartier & Philippe, 1993; Massaroli et al., 2021) that splits the time horizon as multiple regions. However, in our case each region is infinitesimally small, making the Newton step follows equation 9 and parallelizable. The details of the relationship between our method with direct multiple shooting method can be seen in Appendix A.2.

### 3.4 Parallelizing RNN

Recurrent Neural Network (RNN) can be seen as a discretization of ODE. Having the input signal at index $i$ as $\mathbf{x}_i$ and the previous states $\mathbf{y}_{i-1}$, the current states can be written as $\mathbf{y}_i = \mathbf{f}(\mathbf{y}_{i-1}, \mathbf{x}_i, \theta)$. This form can capture the common RNN units, such as LSTM (Hochreiter & Schmidhuber, 1997) and GRU (Cho et al., 2014). Also, the form can be written as equation 1 with $\mathbf{r} = i$, $L[\mathbf{y}] = \mathbf{y}$, $P = 1$ and $\mathbf{s}_1 = 1$. This means that the inverse linear operator can be calculated by solving the equation below, given the initial states $\mathbf{y}_0$,

$$\mathbf{y}_i + \mathbf{G}_i \mathbf{y}_{i-1} = \mathbf{z}_i \iff \mathbf{y}_{1...T} = L_{\mathbf{G}}^{-1}[\mathbf{z}_{1...T}]. \tag{11}$$

Solving the equation above is equivalent to solving equation 9 from the previous subsection. It means that it can be parallelized by using the parallel prefix scan with the defined associative operator in equation 10.

### 3.5 Complexity and limitations

The algorithms for solving non-linear ODE and RNN in parallel are simply repeatedly evaluating equation 3 from an initial guess. However, the equation requires the explicit Jacobian matrix for each sequence element for each shift. If $\mathbf{y}$ has $n$ elements, $L$ sampling points, and $P$ shifted arguments, storing the Jacobian matrices requires $O(n^2 LP)$ memory. Also, the prefix scan for solving the linear differential equation requires matrix multiplication of matrices $\mathbf{G}_p$ at different sampling points. This would introduce $O(n^3 LP)$ time complexity. As the algorithm has $O(n^2)$ memory complexity and $O(n^3)$ time complexity, this algorithm would offer significant acceleration for small number of $n$.

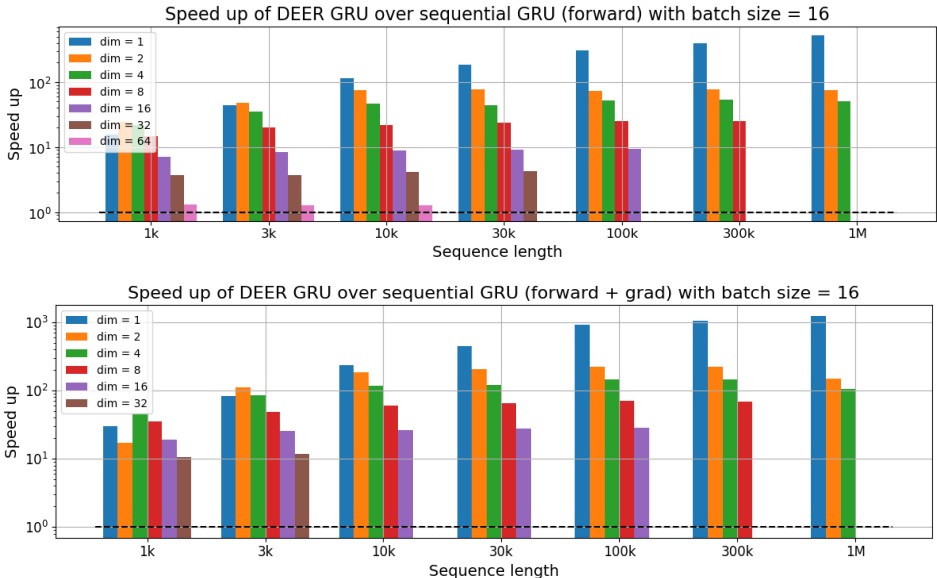

Figure 2: The speed up of GRU calculated using DEER method (this paper) vs commonly-used sequential method on a V100 GPU for (top) forward and (bottom) forward + gradient calculations. The missing data for large number of dimensions and sequence lengths is due to insufficient memory in the DEER method. The bar height represents the mean speed up over 5 different random seeds.

As the proposed method is a realization of Newton's method in Banach space, it has the same limitations as Newton's method. If the starting point is sufficiently far from the solution, the iteration might not converge. This problem might be addressed by using modified Newton's method with convergence guarantee such as Nesterov & Polyak (2006) and Doikov & Nesterov (2023). However, we leave the use of globally-converged Newton's method as the future work.

There is only one additional hyperparameter in our method: the tolerance for convergence. As our method has quadratic convergence, the tolerance value does not have a big effect on the number of iterations required for convergence, as long as it is not too close to the numerical precision value and not too big. In fact, using respectively $10^{-4}$ and $10^{-7}$ for single- and double-precision floating point precisions are enough for our experiments.

Parallelizing ODE and RNN both require parallelizing the evaluation of equations 9 and 11. Those equations can be parallelized using parallel prefix scan of a custom associative operation in equation 10. This is relatively straightforward using JAX's `jax.lax.associative_scan` (Frostig et al., 2018) or TensorFlow's `tfp.math.scan_associative` (Abadi, 2016). However, as of the time of writing this paper, this operation cannot be implemented easily using PyTorch (Paszke et al., 2017). For this reason, we used JAX for our experiments in this paper as well as flax and equinox (Kidger & Garcia, 2021) for the neural networks.

## 4 EXPERIMENTS

### 4.1 PERFORMANCE BENCHMARKING

The first test is to compare the speed on evaluating an RNN using the presented DEER method against the common sequential method. Specifically, we're using an untrained Gated Recurrent Unit (GRU) cell from flax.linen (a JAX neural network framework) using 32-bits floating points random inputs with 16 batch size, various number of dimensions (i.e., $n$), and various number of sequence lengths. The initial guess for DEER is all zeros for all the benchmark runs. The speed up on a V100 GPU obtained by the method presented in this paper is shown in figure 2.

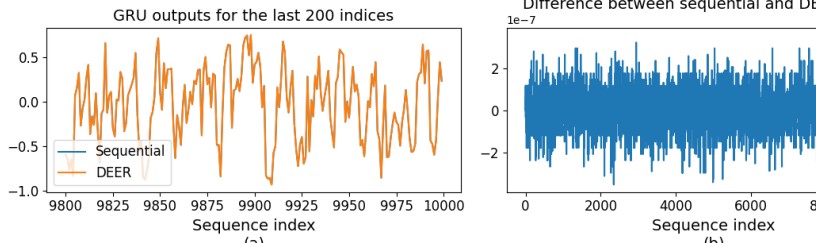

Figure 3: (a) The comparison between the outputs of GRU evaluated with sequential method vs DEER method. The line for sequential method output is almost not visible because overlaid by the output of DEER method. Only the last 200 indices are shown for clarity. (b) The difference between the outputs of sequential and DEER method for the whole 10k sample length.

The figure shows that the largest speed up is achieved on long sequence lengths with small number of dimensions. With 1M sequence length, 16 batch size, and $n = 1$, the sequential evaluation required 8.7 s while DEER method only took 15 ms, which translates to a speed up of over 500. However, the speed up decreases as the number of dimensions increases, where the speed up is only about 25% with 64 dimensions (64 hidden elements in GRU). This is due to the explicit computation of Jacobian matrix and the matrix multiplication that scales to $O(n^3)$.

The speed up for forward + gradient calculations is even greater than the speed up for forward evaluations only. With the same set up, the speed up for 1M sequence length with 1 dimension could be more than 1000 times faster. This is because the backward gradient calculations require only one evaluation of $L_{\mathbf{G}}^{-1}$ in equation 7 as opposed to multiple evaluations in forward calculations.

Table 4 in Appendix C.2 presents the tables of speed up for Figure 2 as well as speed up when using smaller batch sizes. Generally, the speed up increases with smaller batch size, where speed up of above 2600 can be achieved with batch size 2. This means that with more devices, greater speed up can be achieved by having a distributed data parallel, effectively reducing the batch size per device.

Figure 3 shows the comparison between the output of an untrained GRU of 32 hidden elements evaluated using sequential vs DEER method. The input to the GRU is a Gaussian-random tensor with 10k sequence length and 32 dimensions. From figure 3, we see that the output of GRU evaluated using DEER method is almost identical to the output obtained using the sequential method. The small error in figure 3(b) is due to numerical precision limits of the single precision floating point.

## 4.2 LEARNING PHYSICAL SYSTEMS WITH NEURALODE

We test the capability of DEER method in training a deep neural network model using a simple case from physics. Given the positions and velocities as a function of time of a two-body system interacting with gravitational force, we trained Hamiltonian Neural Networks (HNN) (Greydanus et al., 2019) to match the data. There are $n = 8$ states in this case. Note that this set up is different from the original HNN paper where they train the network to match the velocity + acceleration, given position + velocity at every point in time. In this case, only positions and velocities as a function of time are given and the network needs to solve the ODE to make a prediction. We use 10,000 time points sampled uniformly, whereas other works in this area typically only use less than 1,000 sampling points for the training (Matsubara & Yaguchi, 2022; Chen et al., 2019). The details of the setup can be seen in Appendix B.2.

The losses during the training using DEER method vs using RK45 (Atkinson, 1991) are shown in figure 4(a, b). We use the RK45 algorithm from JAX's experimental feature. From the figure, it can be seen that the training can be 11 times faster when using DEER method presented in this paper than using the ordinary ODE solver without significant difference in the validation losses between the two methods. To achieve 55k training steps, DEER method only spent 1 day + 6 hours, while training with RK45 required about 2 weeks of training. The small difference of the validation losses between the two methods potentially comes from different methods in solving the ODE as well as from the numerical precision issue as shown in Figure 3(b).

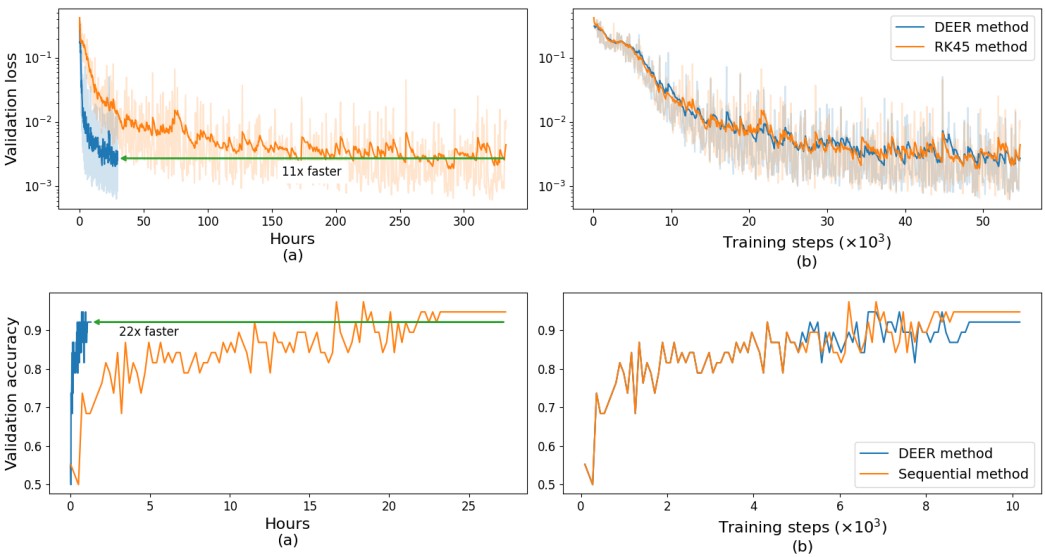

Figure 4: (Top) The validation losses of HNN with NeuralODE training using DEER method (shown in blue) vs RK45 method (in orange) as a function of (a) training hours and (b) training steps. (Bottom) The validation accuracy of RNN training using DEER method (blue) vs the sequential method (orange) as a function of (c) training hours and (d) training steps.

### 4.3 TIME-SERIES CLASSIFICATION WITH RECURRENT NEURAL NETWORK (RNN)

Faster training enabled by DEER method allows us to use classical RNN architectures, such as Gated Recurrent Unit (GRU, Cho et al. (2014)), for problems with long time series. In this subsection, we trained a neural network that consists of GRUs to classify Eigen-Worms dataset (Brown et al., 2013) from UEA (Bagnall et al., 2018) with 17,984 time samples for each entry. The details of the training and architecture can be found in Appendix B.3.

Figure 4(c)-(d) shows the comparison of validation accuracy during the training of the GRU network using DEER method vs the common sequential method. From the figure, we can see that the validation accuracy plot when using DEER is similar to the validation plot obtained with the sequential method. The difference in the validation accuracy might be due to numerical precision issue as shown in Figure 3(b) that

| Model | Accuracy (%) |
|---|---|
| ODE-RNN (folded), step: 128 | $47.9 \pm 5.3$ |
| NCDE, step: 4 | $66.7 \pm 11.8$ |
| NRDE (depth 3), step: 32 | $75.2 \pm 3.0$ |
| NRDE (depth 2), step: 128 | $76.1 \pm 5.9$ |
| NRDE (depth 2), step: 4 | $\mathbf{83.8 \pm 3.0}$ |
| UnICORNN (2 layers) | $\mathbf{90.3 \pm 3.0}$ |
| LEM | $\mathbf{92.3 \pm 1.8}$ |
| GRU (from this paper) | $\mathbf{88.0 \pm 4.4}$ |
| LEM (our reproducibility attempt) | $\mathbf{92.3 \pm 2.1}$ |

Table 1: The classification accuracy of EigenWorms dataset for various methods, including folded ODE-RNN (Rubanova et al., 2019), Neural CDE (Kidger et al., 2020), Neural RDE (Morrill et al., 2021), UnICORNN (Rusch & Mishra, 2021), LEM (Rusch et al., 2021), and GRU. The mean and standard deviations of the accuracy were obtained from 3 times repetition with different seeds. The numbers of non-GRU methods were obtained from Morrill et al. (2021) and Rusch et al. (2021).

is accumulated at long sequence. However, the training using DEER is up to 22 times faster than the training using the sequential method. What would have taken more than 1 day to train using the common sequential method, only takes 1 hour using DEER.

The classification results of the test dataset from EigenWorms for various methods are shown in Table 1. As we can see from the table, the network with GRUs can be as competitive as more

modern methods (Morrill et al., 2021; Rusch & Mishra, 2021; Rusch et al., 2021) for this long time series dataset. The long training time of GRUs using the sequential method could be the main factor hindering the trial-and-error process of exploring GRU architectures for this dataset with long sequences. However, our DEER method enables much faster training of GRUs, facilitating iterative experimentation to identify optimal GRU architectures and hyperparameters.

To demonstrate the applicability of DEER to other architecture, we performed an experiment to reproduce the results from LEM (Rusch et al., 2021). The reimplementation of LEM from PyTorch to JAX makes the reproducibility attempt not straightforward. Moreover, as the EigenWorms dataset is very small, slight changes could change the results quite significantly. With DEER, we can run more experiments than using sequential methods, allowing us to tune hyperparameters faster. Our attempt has produced similar results to the original attempt at Rusch et al. (2021). With DEER, one epoch can be done in 18 seconds while the sequential method could take about 116 seconds.

## 4.4 SEQUENCE CLASSIFICATION WITH MULTIPLE HEADS RNN

One way to avoid unfavorable scaling with respect to $n$ in DEER is to use multi-head recurrent unit. For example, instead of having a unit with $n = 256$ channels, we split the unit into 32 heads with each head has $n = 8$ channels. Each head can also have different strides with exponentially increasing values, for example, the first head has stride $2^0$, second head has stride $2^1$, and so on. The use of multiple heads is inline with the idea from state-spaces (Gu et al., 2021b;a; 2022; Smith et al., 2022) where states are grouped and each group has different length scale.

We performed an experiment using the multi-head GRU on a standard sequential CIFAR-10 dataset. The details of the experimental setup can be found in Appendix B.4. Running an experiment with DEER using multi-head GRU is about 3 times faster than using sequential method. The speed up is not as big as in the previous subsections because of shorter effective sequence lengths due to larger strides. However, the speed up still allows us to run multiple experiments to find a good configuration.

| Model | Accuracy |
|---|---|
| *State-space or linear recurrent* | |
| LSSL (Gu et al., 2021b) | 84.65% |
| S4 (Gu et al., 2021a) | 91.80% |
| Liquid-S4 (Hasani et al., 2022) | 92.02% |
| LRU (Orvieto et al., 2023) | 89.0% |
| *Convolution* | |
| TrellisNet (Bai et al., 2018) | 73.42% |
| FlexConv (Romero et al., 2021) | 80.82% |
| MultiresNet (Shi et al., 2023) | **93.15%** |
| *Non-linear recurrent* | |
| r-LSTM (Trinh et al., 2018) | 72.2% |
| UR-GRU (Gu et al., 2020) | 74.4% |
| Multi-head GRU (ours) | 90.25% |

Table 2: The classification test dataset accuracy of sequential CIFAR-10 dataset for various methods, grouped according to their architecture classes. Underlined scores are the best score for each architecture class.

Table 2 shows that multi-head GRU can achieve 89.35% test accuracy. Although the result is not the state-of-the-art, this is the best result on sequential CIFAR-10 using non-linear recurrent units. We show that a simple modification of GRU and the capability to run more experiments enabled by DEER has uncover the potential of classical recurrent units such as GRU in competing with more modern architectures. Please note that our aim for this paper is not to focus on new architectures, but to focus on a new computational method that enables faster experimentation on non-linear sequential models. Finding new non-linear sequential models enabled by DEER is for the future work.

## 5 CONCLUSION

We introduced a method to parallelize the evaluation and training of inherently sequential models, such as ODE and RNN. Evaluations using our approach can be up to 3 orders of magnitude faster than traditional sequential methods when dealing with long sequences. When training sequential models with reasonable settings, our method can achieve over a 10-fold speed increase without significantly altering the results. By having a technique to accelerate the training and evaluation of sequential models, we anticipate a hastened pace of research in this domain, potentially catalyzing the emergence of novel and interesting non-linear sequential models in the future.

## REPRODUCIBILITY STATEMENT

The code required for reproducing the algorithm and results in this paper can be found in https://github.com/machine-discovery/deer/.

## ACKNOWLEDGEMENT

We acknowledge the support from Google Cloud start up program for their cloud computing credit that being used in large part of this research.

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

# A   THEORETICAL RESULTS

## A.1   RELATION TO NEWTON'S METHOD

The main method presented in this paper is basically a realization of Newton's method in Banach space. Finding the root of functional $\mathbf{q}(\mathbf{y}) = \mathbf{0}$ can be done by performing the Newton's method iteration $\mathbf{y}^{(i+1)} = \mathbf{y}^{(i)} - [D\mathbf{q}(\mathbf{y}^{(i)})]^{-1}\mathbf{q}(\mathbf{y}^{(i)})$ where $D\mathbf{q}$ is the Fréchet derivative of $\mathbf{q}$. In equation 1, the function $\mathbf{q}(\mathbf{y})$ is given by $\mathbf{q}(\mathbf{y}(\mathbf{r})) = L[\mathbf{y}(\mathbf{r})] - \mathbf{f}(\mathbf{y}(\mathbf{r} - \mathbf{s}_1), ..., \mathbf{x}(\mathbf{r}), \theta)$. Therefore, the Fréchet derivative of functional $\mathbf{q}$ operated on a function $\mathbf{h}(\mathbf{r})$ is given by

$$D\mathbf{q}(\mathbf{y})[\mathbf{h}(\mathbf{r})] = L[\mathbf{h}(\mathbf{r})] - \sum_{p=1}^{P} \partial_p \mathbf{f} \mathbf{h}(\mathbf{r} - \mathbf{s}_p). \tag{12}$$

Therefore, the Newton's method iteration to find the root of functional $\mathbf{q}(\mathbf{y}) = \mathbf{0}$ is

$$\mathbf{y}^{(i+1)} = \mathbf{y}^{(i)} - (D\mathbf{q}(\mathbf{y}))^{-1} \left[ L[\mathbf{y}^{(i)}] - \mathbf{f}(\mathbf{y}^{(i)}(\mathbf{r} - \mathbf{s}_1), ..., \mathbf{x}(\mathbf{r}), \theta) \right]. \tag{13}$$

The inverse Fréchet derivative terms can be computed by solving the equation below for $\mathbf{y}^{(i)} - \mathbf{y}^{(i+1)}$,

$$D\mathbf{q}(\mathbf{y}) \left[ \mathbf{y}^{(i)} - \mathbf{y}^{(i+1)} \right] = L[\mathbf{y}^{(i)}] - \mathbf{f}(\mathbf{y}^{(i)}(\mathbf{r} - \mathbf{s}_1), ..., \mathbf{x}(\mathbf{r}), \theta) \tag{14}$$

Expanding $D\mathbf{q}(\mathbf{y})$ from equation 12 on $\mathbf{y}^{(i)}$, we obtain

$$D\mathbf{q}(\mathbf{y})[\mathbf{y}^{(i+1)}] = \mathbf{f}(\mathbf{y}^{(i)}(\mathbf{r} - \mathbf{s}_1), ..., \mathbf{x}(\mathbf{r}), \theta) - \sum_{p=1}^{P} \partial_p \mathbf{f} \mathbf{y}^{(i)}(\mathbf{r} - \mathbf{s}_p) \tag{15}$$

Solving for $\mathbf{y}^{(i+1)}$ on the equation above will produce the equation 3, showing that equation 3 is just a realization of Newton's method in solving equation 1.

## A.2   RELATION TO DIRECT MULTIPLE SHOOTING

One popular way to parallelize an ODE solver is to perform direct multiple shooting (MS) method (Chartier & Philippe, 1993; Massaroli et al., 2021). The following parts will follow Massaroli et al. (2021) very closely. Consider an initial value problem with $d\mathbf{y}/dt = \mathbf{f}(\mathbf{y}, t)$ for $t \in [0, T]$ with $\mathbf{y}(0) = \mathbf{y}_0$. With MS, we first split the time horizon into $N$ multiple regions with $0 = t_0 < t_1 < t_2 < ... < t_N = T$, have a guess on initial values at each region, $\mathbf{y}(t_i) = \mathbf{b}_i$, then apply the ODE solver for each region. After applying the ODE, the guessed values $\mathbf{b}_i$ can then be updated to match the results of ODE from the previous region. Denote a function $\phi(\mathbf{b}_i, t_i, t_{i+1})$ as the value of $\mathbf{y}$ at $t_{i+1}$ after solving the initial value problem with initial value $\mathbf{y}(t_i) = \mathbf{b}_i$. Evaluating this function $\phi$ requires solving an ODE from $t_i$ to $t_{i+1}$. With the function above, we can write the constraints to be solved, $\mathbf{b}_{i+1} = \phi(\mathbf{b}_i, t_i, t_{i+1})$ with $\mathbf{b}_0 = \mathbf{y}_0$.

The works by Chartier & Philippe (1993) and Massaroli et al. (2021) update $\mathbf{b}_i$ by applying Newton method. Specifically, at $k$-th iteration with values of $\mathbf{b}_i^{(k)}$ are available for all $i$-s, the values at the next iteration can be determined by (Massaroli et al., 2021)

$$\mathbf{b}_{i+1}^{(k+1)} = \phi(\mathbf{b}_i^{(k)}, t_i, t_{i+1}) + \frac{\partial \phi}{\partial \mathbf{b}}(\mathbf{b}_i^{(k)}, t_i, t_{i+1}) \left( \mathbf{b}_i^{(k+1)} - \mathbf{b}_i^{(k)} \right); \quad \mathbf{b}_0^{(k+1)} = \mathbf{y}_0. \tag{16}$$

The term $\mathbf{b}_i^{(k+1)}$ makes the equation above needs to be evaluated sequentially, in addition to computing the function $\phi$ that requires solving an ODE.

If we assume there are large number of regions, $N \rightarrow \infty$ and $\Delta t_i = t_{i+1} - t_i \rightarrow 0$, then the MS method becomes equivalent to the DEER realization on solving the ODE. In this limit, we can write

$$\phi(\mathbf{b}_i, t_i, t_{i+1}) = \mathbf{b}_i + \mathbf{f}(\mathbf{b}_i, t_i)\Delta t_i \text{ and } \frac{\partial \phi}{\partial \mathbf{b}} = \mathbf{I} + \frac{\partial \mathbf{f}}{\partial \mathbf{b}}(\mathbf{b}_i, t_i), \tag{17}$$

with $\Delta t_i = t_{i+1} - t_i$ and $\mathbf{I}$ an identity matrix. Substituting equation 17 to equation 16, we obtain

$$\mathbf{b}_{i+1}^{(k+1)} = \left( \mathbf{I} + \frac{\partial \mathbf{f}}{\partial \mathbf{b}}(\mathbf{b}_i^{(k)}, t_i)\Delta t_i \right) \mathbf{b}_i^{(k+1)} + \left[ \mathbf{f}(\mathbf{b}_i^{(k)}, t_i) - \frac{\partial \mathbf{f}}{\partial \mathbf{b}}(\mathbf{b}_i^{(k)}, t_i)\mathbf{b}_i^{(k)} \right] \Delta t_i. \tag{18}$$

By re-arranging the equation above, we get

$$\left(\frac{\mathbf{b}_{i+1}^{(k+1)} - \mathbf{b}_i^{(k+1)}}{\Delta t_i}\right) - \frac{\partial \mathbf{f}}{\partial \mathbf{b}}(\mathbf{b}_i^{(k)}, t_i)\mathbf{b}_i^{(k+1)} = \mathbf{f}(\mathbf{b}_i^{(k)}, t_i) - \frac{\partial \mathbf{f}}{\partial \mathbf{b}}(\mathbf{b}_i^{(k)}, t_i)\mathbf{b}_i^{(k)}. \tag{19}$$

With the limit $\Delta t_i \to 0$ and $N \to \infty$, the term $\mathbf{b}_i$ can be written as $\mathbf{y}(t_i)$ and $(\mathbf{b}_{i+1}^{(k+1)} - \mathbf{b}_i^{(k)})/\Delta t_i$ equals to $d\mathbf{y}^{(k+1)}/dt$. By rewriting the equation above in $\mathbf{y}$, we obtain

$$\frac{d\mathbf{y}^{(k+1)}}{dt} - \frac{\partial \mathbf{f}}{\partial \mathbf{y}}(\mathbf{y}^{(k)}, t)\mathbf{y}^{(k+1)} = \mathbf{f}(\mathbf{y}^{(k)}, t) - \frac{\partial \mathbf{f}}{\partial \mathbf{y}}(\mathbf{y}^{(k)}, t)\mathbf{y}^{(k)}. \tag{20}$$

With a little bit of algebra, one can show that the equation above is equivalent to equations 3 and 8. By having a large number of regions, we can approximate $\phi$ as in equation 17 that can be evaluated very quickly.

### A.3 PROOF OF QUADRATIC CONVERGENCE OF DEER ITERATION

This proof is inspired by standard results on the order of convergence of Newton's method and follows the argument of Kelley (1995), §§4 and 5.1. Throughout, we will make repeated use of the Cauchy-Schwarz (CS) and triangle (T) inequalities.

We employ the standard Euclidean norm on $\mathbb{R}^n$,

$$\|\mathbf{y}\| := \sqrt{\mathbf{y} \cdot \mathbf{y}} = \sqrt{y_1^2 + \cdots + y_n^2}, \ \mathbf{y} = (y_1, \ldots, y_n)^T \in \mathbb{R}^n, \tag{21}$$

and the induced operator norm on $\mathbb{R}^{n \times n}$,

$$\|\mathbf{A}\| := \sup\{\|\mathbf{A}\mathbf{y}\|, \ \|\mathbf{y}\| = 1, \ \mathbf{A} \in \mathbb{R}^{n \times n}, \ \mathbf{y} \in \mathbb{R}^n\}. \tag{22}$$

Note that the Fréchet derivative $D\mathbf{q}(\mathbf{y}) \in \mathbb{R}^{n \times n}$.

We denote the value of $\mathbf{y}$ at the $i$th iteration as $\mathbf{y}^{(i)}(\mathbf{r}) = \mathbf{y}^*(\mathbf{r}) + \delta\mathbf{y}^{(i)}(\mathbf{r})$, where $\mathbf{y}^*$ exactly satisfies equation 3. Following on from A.1, we assume that $D\mathbf{q}(\mathbf{y})$ is Lipschitz continuous with Lipschitz constant $\gamma$, i.e.:

$$\|D\mathbf{q}(\mathbf{y}) - D\mathbf{q}(\mathbf{z})\| \le \gamma\|\mathbf{y} - \mathbf{z}\| \quad \forall \, \mathbf{y}, \mathbf{z} \in \mathbb{R}^n. \tag{23}$$

Consider the operators $\mathbf{A} = D\mathbf{q}(\mathbf{y})$ and $\mathbf{B} = D\mathbf{q}(\mathbf{y}^*)$. We assume that both are invertible and that $\|\mathbf{I} - \mathbf{B}^{-1}\mathbf{A}\| < 1$:

$$\begin{aligned}
\mathbf{A}^{-1}\mathbf{B} &= \left(\mathbf{I} - \left(\mathbf{I} - \mathbf{B}^{-1}\mathbf{A}\right)\right)^{-1} \\
&= \sum_{n=0}^{\infty} \left(\mathbf{I} - \mathbf{B}^{-1}\mathbf{A}\right)^n \qquad \text{(Neumann series for matrix inverse)}.
\end{aligned} \tag{24}$$

Post-multiplying by $\mathbf{B}^{-1}$ and taking norms:

$$\begin{aligned}
\|\mathbf{A}^{-1}\| &= \left\|\left(\sum_{n=0}^{\infty}\left(\mathbf{I} - \mathbf{B}^{-1}\mathbf{A}\right)^n\right)\mathbf{B}^{-1}\right\| \\
&\overset{(CS)}{\le} \left\|\sum_{n=0}^{\infty}\left(\mathbf{I} - \mathbf{B}^{-1}\mathbf{A}\right)^n\right\|\|\mathbf{B}^{-1}\| \\
&\overset{(T)}{\le} \left(\sum_{n=0}^{\infty}\left\|\left(\mathbf{I} - \mathbf{B}^{-1}\mathbf{A}\right)^n\right\|\right)\|\mathbf{B}^{-1}\| \\
&\overset{(CS)}{\le} \left(\sum_{n=0}^{\infty}\|\mathbf{I} - \mathbf{B}^{-1}\mathbf{A}\|^n\right)\|\mathbf{B}^{-1}\| \\
&= \frac{\|\mathbf{B}^{-1}\|}{1 - \|\mathbf{I} - \mathbf{B}^{-1}\mathbf{A}\|} \\
\|[D\mathbf{q}(\mathbf{y})]^{-1}\| &\le \frac{\|[D\mathbf{q}(\mathbf{y}^*)]^{-1}\|}{1 - \|\mathbf{I} - [D\mathbf{q}(\mathbf{y}^*)]^{-1}D\mathbf{q}(\mathbf{y})\|}.
\end{aligned} \tag{25}$$

We denote the ball of radius $\epsilon$ as $\mathcal{B}(\epsilon) = \{\mathbf{y} \mid \|\delta\mathbf{y}\| < \epsilon\}$, where $\delta\mathbf{y} = \mathbf{y} - \mathbf{y}^*$.

$$
\begin{aligned}
\left\|\mathbf{I} - [D\mathbf{q}(\mathbf{y}^*)]^{-1} D\mathbf{q}(\mathbf{y})\right\| &= \left\|[D\mathbf{q}(\mathbf{y}^*)]^{-1} \left(D\mathbf{q}(\mathbf{y}^*) - D\mathbf{q}(\mathbf{y})\right)\right\| \\
&\overset{(CS)}{\leq} \left\|[D\mathbf{q}(\mathbf{y}^*)]^{-1}\right\| \left\|D\mathbf{q}(\mathbf{y}^*) - D\mathbf{q}(\mathbf{y})\right\| \\
&\overset{(23)}{\leq} \left\|[D\mathbf{q}(\mathbf{y}^*)]^{-1}\right\| \gamma \|\mathbf{y}^* - \mathbf{y}\| \\
&= \gamma \|\delta\mathbf{y}\| \left\|[D\mathbf{q}(\mathbf{y}^*)]^{-1}\right\|
\end{aligned}
\tag{26}
$$

Let $0 < \epsilon < \left(2\gamma\left\|[D\mathbf{q}(\mathbf{y}^*)]^{-1}\right\|\right)^{-1}$. Then $\forall\, \mathbf{y} \in \mathcal{B}(\epsilon)$:

$$
\begin{aligned}
\left\|\mathbf{I} - [D\mathbf{q}(\mathbf{y}^*)]^{-1} D\mathbf{q}(\mathbf{y})\right\| &\leq \gamma \|\delta\mathbf{y}\| \left\|[D\mathbf{q}(\mathbf{y}^*)]^{-1}\right\| \\
&\leq \gamma\epsilon \left\|[D\mathbf{q}(\mathbf{y}^*)]^{-1}\right\| \\
&< \frac{1}{2}.
\end{aligned}
\tag{27}
$$

This means that

$$
\left\|[D\mathbf{q}(\mathbf{y})]^{-1}\right\| \leq 2\left\|[D\mathbf{q}(\mathbf{y}^*)]^{-1}\right\|.
\tag{28}
$$

The fundamental theorem of calculus can be expressed as:

$$
\begin{aligned}
\mathbf{q}(\mathbf{y}) - \mathbf{q}(\mathbf{y}^*) &= \int_0^1 D\mathbf{q}\left(\mathbf{y}^* + \lambda\left(\mathbf{y} - \mathbf{y}^*\right)\right)\left(\mathbf{y} - \mathbf{y}^*\right) d\lambda \\
\mathbf{q}(\mathbf{y}) &= \int_0^1 D\mathbf{q}\left(\mathbf{y}^* + \lambda\delta\mathbf{y}\right)\delta\mathbf{y}\, d\lambda.
\end{aligned}
\tag{29}
$$

Therefore,

$$
\begin{aligned}
\delta\mathbf{y}^{(i+1)} &= \delta\mathbf{y}^{(i)} - [D\mathbf{q}(\mathbf{y}^{(i)})]^{-1}\mathbf{q}(\mathbf{y}^{(i)}) \\
&= \delta\mathbf{y}^{(i)} - [D\mathbf{q}(\mathbf{y}^{(i)})]^{-1} \int_0^1 D\mathbf{q}\left(\mathbf{y}^* + \lambda\delta\mathbf{y}^{(i)}\right)\delta\mathbf{y}^{(i)} d\lambda \\
&= [D\mathbf{q}(\mathbf{y}^{(i)})]^{-1} \int_0^1 \left(D\mathbf{q}(\mathbf{y}^{(i)}) - D\mathbf{q}\left(\mathbf{y}^* + \lambda\delta\mathbf{y}^{(i)}\right)\right)\delta\mathbf{y}^{(i)} d\lambda.
\end{aligned}
\tag{30}
$$

Let $\mathbf{y}^{(i)} \in \mathcal{B}(\epsilon)$. Then:

$$
\begin{aligned}
\left\|\delta\mathbf{y}^{(i+1)}\right\| &= \left\|[D\mathbf{q}(\mathbf{y}^{(i)})]^{-1} \int_0^1 \left(D\mathbf{q}(\mathbf{y}^{(i)}) - D\mathbf{q}\left(\mathbf{y}^* + \lambda\delta\mathbf{y}^{(i)}\right)\right)\delta\mathbf{y}^{(i)} d\lambda\right\| \\
&\overset{(CS)}{\leq} \left\|[D\mathbf{q}(\mathbf{y}^{(i)})]^{-1}\right\| \left\|\int_0^1 \left(D\mathbf{q}(\mathbf{y}^{(i)}) - D\mathbf{q}\left(\mathbf{y}^* + \lambda\delta\mathbf{y}^{(i)}\right)\right)\delta\mathbf{y}^{(i)} d\lambda\right\| \\
&\overset{(28)}{\leq} 2\left\|[D\mathbf{q}(\mathbf{y}^*)]^{-1}\right\| \left\|\int_0^1 \left(D\mathbf{q}(\mathbf{y}^{(i)}) - D\mathbf{q}\left(\mathbf{y}^* + \lambda\delta\mathbf{y}^{(i)}\right)\right)\delta\mathbf{y}^{(i)} d\lambda\right\| \\
&\overset{(T)}{\leq} 2\left\|[D\mathbf{q}(\mathbf{y}^*)]^{-1}\right\| \int_0^1 \left\|\left(D\mathbf{q}(\mathbf{y}^{(i)}) - D\mathbf{q}\left(\mathbf{y}^* + \lambda\delta\mathbf{y}^{(i)}\right)\right)\delta\mathbf{y}^{(i)}\right\| d\lambda \\
&\overset{(CS)}{\leq} 2\left\|[D\mathbf{q}(\mathbf{y}^*)]^{-1}\right\| \int_0^1 \left\|D\mathbf{q}(\mathbf{y}^{(i)}) - D\mathbf{q}\left(\mathbf{y}^* + \lambda\delta\mathbf{y}^{(i)}\right)\right\| d\lambda \left\|\delta\mathbf{y}^{(i)}\right\| \\
&\overset{(23)}{\leq} 2\left\|[D\mathbf{q}(\mathbf{y}^*)]^{-1}\right\| \int_0^1 \gamma\left\|\mathbf{y}^{(i)} - \left(\mathbf{y}^* + \lambda\delta\mathbf{y}^{(i)}\right)\right\| d\lambda \left\|\delta\mathbf{y}^{(i)}\right\| \\
&= 2\left\|[D\mathbf{q}(\mathbf{y}^*)]^{-1}\right\| \int_0^1 \gamma\left\|(1-\lambda)\delta\mathbf{y}^{(i)}\right\| d\lambda \left\|\delta\mathbf{y}^{(i)}\right\| \\
&\overset{(CS)}{\leq} 2\left\|[D\mathbf{q}(\mathbf{y}^*)]^{-1}\right\| \int_0^1 \gamma\|1-\lambda\| d\lambda \left\|\delta\mathbf{y}^{(i)}\right\|^2 \\
&= 2\left\|[D\mathbf{q}(\mathbf{y}^*)]^{-1}\right\| \frac{\gamma}{2} \left\|\delta\mathbf{y}^{(i)}\right\|^2 \\
&= \gamma\left\|[D\mathbf{q}(\mathbf{y}^*)]^{-1}\right\| \left\|\delta\mathbf{y}^{(i)}\right\|^2.
\end{aligned}
\tag{31}
$$

So $\left\|\delta\mathbf{y}^{(i+1)}\right\|$ is bounded by $\left\|\delta\mathbf{y}^{(i)}\right\|^2$ with constant $\gamma\left\|[D\mathbf{q}(\mathbf{y}^*)]^{-1}\right\|$. For quadratic *convergence*, the norms of successive errors must decrease. If we shrink $\epsilon$ such that $\gamma\epsilon\left\|[D\mathbf{q}(\mathbf{y}^*)]^{-1}\right\| < 1$, then:

$$
\begin{aligned}
\left\|\delta\mathbf{y}^{(i+1)}\right\| &\leq \gamma\left\|[D\mathbf{q}(\mathbf{y}^*)]^{-1}\right\|\left\|\delta\mathbf{y}^{(i)}\right\|^2 \\
&< \gamma\epsilon\left\|[D\mathbf{q}(\mathbf{y}^*)]^{-1}\right\|\left\|\delta\mathbf{y}^{(i)}\right\| \\
&< \left\|\delta\mathbf{y}^{(i)}\right\|
\end{aligned}
\tag{32}
$$

and the sequence therefore converges quadratically to $\mathbf{y}^*$. $\qquad\square$

### A.4 APPLICATION TO PARTIAL DIFFERENTIAL EQUATIONS (PDES)

We have mentioned that equation 1 can be applied to Partial Differential Equations (PDEs). As this paper is concerning about Ordinary Differential Equations (ODEs) and discrete difference equation, we only present in this appendix the example of equation 1 in representing a simple PDE. Let's rewrite the equation 1 as a reminder,

$$
L[\mathbf{y}(\mathbf{r})] = \mathbf{f}\left(\mathbf{y}(\mathbf{r}-\mathbf{s}_1), ..., \mathbf{y}(\mathbf{r}-\mathbf{s}_P), \mathbf{x}(\mathbf{r}), \theta\right),
\tag{33}
$$

where $L[\cdot]$ is the linear operator, $\mathbf{f}$ is the non-linear function, $\mathbf{y}$ is the signal of interest, $\mathbf{x}$ is the external signal, $\mathbf{r}$ is the coordinate, $\theta$ is the parameters of $\mathbf{f}$, and $\mathbf{s}_i$ is the shifted location.

We will take the Burgers' equation as an example. Burgers' equation can be written as

$$
\frac{\partial u}{\partial t} + \frac{1}{2}\frac{\partial(u^2)}{\partial x} - \nu\frac{\partial^2 u}{\partial x^2} = 0
\tag{34}
$$

where $u$ is a function of $x$ and $t$, $u(x,t)$. If we define $w = u^2$, we can write the Burgers' equation to be

$$
\begin{pmatrix} \frac{\partial}{\partial t} - v\frac{\partial^2}{\partial x^2} & \frac{1}{2}\frac{\partial}{\partial x} \\ 0 & 1 \end{pmatrix} \begin{pmatrix} u \\ w \end{pmatrix} = \begin{pmatrix} 0 \\ u^2 \end{pmatrix}.
\tag{35}
$$

From the equation above, we can define the signal of interest $\mathbf{y} = (u, w)^T$, the coordinates $\mathbf{r} = (x, t)^T$, the non-linear function $\mathbf{f} = (0, u^2)^T$, and the linear operator

$$
L = \begin{pmatrix} \frac{\partial}{\partial t} - v\frac{\partial^2}{\partial x^2} & \frac{1}{2}\frac{\partial}{\partial x} \\ 0 & 1 \end{pmatrix}.
\tag{36}
$$

Given the relations between the Burgers' equation and equation 1, we can analytically compute the $\mathbf{G}$ matrix and the argument of $L_{\mathbf{G}}^{-1}$,

$$
\mathbf{G} = -\frac{\partial\mathbf{f}}{\partial\mathbf{y}} = \begin{pmatrix} 0 & 0 \\ -2u & 0 \end{pmatrix}
\tag{37}
$$

$$
\mathbf{h} = \mathbf{f} + \mathbf{G}\mathbf{y} = \begin{pmatrix} 0 \\ -u^2 \end{pmatrix}.
\tag{38}
$$

Therefore, the linear operation that needs to be solved, $L_{\mathbf{G}}^{-1}[\mathbf{h}]$, is

$$
\frac{\partial u}{\partial t} + \frac{1}{2}\frac{\partial w}{\partial x} - \nu\frac{\partial^2 u}{\partial x^2} = 0
\tag{39}
$$

$$
g_{21}(x,t)u + w = h_2(x,t),
\tag{40}
$$

given $h_2(x,t) = -u^2$ and $g_{21}(x,t) = -2u$ with $u$ from the previous iteration. The equations above can still be further simplified into

$$
\left[\frac{\partial}{\partial t} - \frac{g_{21}(x,t)}{2}\frac{\partial}{\partial x} - \frac{1}{2}\frac{\partial g_{21}}{\partial x}(x,t) - \nu\frac{\partial^2}{\partial x^2}\right]u = -\frac{1}{2}\frac{\partial h_2}{\partial x}(x,t).
\tag{41}
$$

The equation above can be repeatedly evaluated until the convergence is achieved.

## A.5  ERROR BOUND ON TAKING THE MIDPOINTS IN ODE

The exact analytical solution to the ODE

$$\frac{d\mathbf{y}}{dt}(t) + \mathbf{G}(t)\mathbf{y}(t) = \mathbf{z}(t) \tag{42}$$

is

$$\mathbf{y}(t) = e^{-\mathbf{M}(t)}\left[\mathbf{y}(0) + \int_0^t e^{\mathbf{M}(\tau)}\mathbf{z}(\tau)d\tau\right], \tag{43}$$

where

$$\mathbf{M}(t) = \int_0^t \mathbf{G}(\lambda)d\lambda. \tag{44}$$

Equation 43 is the form that equation 3 takes in the ODE case. The approximation we wish to take involves inserting $\mathbf{G}(t) = \frac{1}{2}(\mathbf{G}(t_i) + \mathbf{G}(t_{i+1}))$ and $\mathbf{z}(t) = \frac{1}{2}(\mathbf{z}(t_i) + \mathbf{z}(t_{i+1}))$ into 43, which yields equation 9. In order to find the local truncation error when applying this method, we need to find and compare the Taylor expansions of equations 43 and 9.

We define $\square_0^{(n)} = \frac{d^n\square}{dt^n}\big|_{t=0}$ so that, for instance, $\mathbf{G}_0 = \mathbf{G}(0), \mathbf{G}_0' = \frac{d\mathbf{G}}{dt}\big|_{t=0}, \mathbf{G}_0'' = \frac{d^2\mathbf{G}}{dt^2}\big|_{t=0}$ and $\mathbf{G}_0^{(3)} = \frac{d^3\mathbf{G}}{dt^3}\big|_{t=0}$. We can then write the Taylor expansion of $\mathbf{M}(t)$ as:

$$\begin{aligned}
\mathbf{M}(t) &= \int_0^t \mathbf{G}(\lambda)d\lambda \\
&= \int_0^t \left(\mathbf{G}_0 + \mathbf{G}_0'\lambda + \frac{1}{2}\mathbf{G}_0''\lambda^2 + \mathcal{O}(\lambda^3)\right)d\lambda \\
&= \mathbf{G}_0 t + \frac{1}{2}\mathbf{G}_0' t^2 + \frac{1}{6}\mathbf{G}_0'' t^3 + \mathcal{O}(t^4),
\end{aligned} \tag{45}$$

whence we immediately get

$$\mathbf{M}_0^{(n)} = \mathbf{G}_0^{(n-1)}, \tag{46}$$

as expected from the Leibniz integral rule. Therefore:

$$\begin{aligned}
e^{\pm\mathbf{M}(t)} &= \mathbf{I} \pm \mathbf{M}(t) + \frac{1}{2}\mathbf{M}(t)^2 \pm \frac{1}{6}\mathbf{M}(t)^3 + \mathcal{O}(\mathbf{M}(t)^4) \\
&= \mathbf{I} \pm \mathbf{G}_0 t + \frac{1}{2}\left(\mathbf{G}_0^2 \pm \mathbf{G}_0'\right)t^2 \pm \frac{1}{12}\left(2\mathbf{G}_0^3 \pm 3\mathbf{G}_0\mathbf{G}_0' \pm 3\mathbf{G}_0'\mathbf{G}_0 + 2\mathbf{G}_0''\right)t^3 + \mathcal{O}(t^4) \\
&\equiv \mathbf{I} \pm \mathbf{G}_0 t + \mathbf{A}_\pm t^2 \pm \mathbf{B}_\pm t^3 + \mathcal{O}(t^4),
\end{aligned} \tag{47}$$

where for ease of notation we have introduced the new variables $\mathbf{A}_\pm = \frac{1}{2}\left(\mathbf{G}_0^2 \pm \mathbf{G}_0'\right)t^2$ and $\mathbf{B}_\pm = \frac{1}{12}\left(2\mathbf{G}_0^3 \pm 3\mathbf{G}_0\mathbf{G}_0' \pm 3\mathbf{G}_0'\mathbf{G}_0 + 2\mathbf{G}_0''\right)$. As the ODE in section 3 is originally cast in the form $\frac{d\mathbf{y}}{dt} = \mathbf{f}(\mathbf{y}(t), \mathbf{x}(t), \theta)$, the right hand side of 42 is

$$\mathbf{z}(t) = \mathbf{f}(\mathbf{y}(t), \mathbf{x}(t), \theta) + \mathbf{G}(t)\mathbf{y}(t), \tag{48}$$

which can be expressed using the Taylor expansions of $\mathbf{f}$, $\mathbf{G}$, and $\mathbf{y}$ as:

$$\mathbf{z}(t) = \mathbf{f}_0 + \mathbf{G}_0\mathbf{y}_0 + (\mathbf{f}_0' + \mathbf{G}_0\mathbf{y}_0' + \mathbf{G}_0'\mathbf{y}_0)t + \frac{1}{2}\left(\mathbf{f}_0'' + \mathbf{G}_0\mathbf{y}_0'' + 2\mathbf{G}_0'\mathbf{y}_0' + \mathbf{G}_0''\mathbf{y}_0\right)t^2 + \mathcal{O}(t^3). \tag{49}$$

Then,

$$
\begin{aligned}
\int_0^t e^{\mathbf{M}(\tau)}\mathbf{z}(\tau)d\tau &= \int_0^t \left(\mathbf{I} + \mathbf{G}_0\tau + \mathbf{A}_+\tau^2 + \mathcal{O}(\tau^3)\right)\Big(\mathbf{f}_0 + \mathbf{G}_0\mathbf{y}_0 + (\mathbf{f}_0' + \mathbf{G}_0\mathbf{y}_0' + \mathbf{G}_0'\mathbf{y}_0)\,\tau \\
&\qquad + \frac{1}{2}\left(\mathbf{f}_0'' + \mathbf{G}_0\mathbf{y}_0'' + 2\mathbf{G}_0'\mathbf{y}_0' + \mathbf{G}_0''\mathbf{y}_0\right)\tau^2 + \mathcal{O}(\tau^3)\Big)d\tau \\
&= \left(\mathbf{f}_0 + \mathbf{G}_0\mathbf{y}_0\right)t + \frac{1}{2}\left(\mathbf{f}_0' + \mathbf{G}_0\mathbf{y}_0' + \mathbf{G}_0'\mathbf{y}_0 + \mathbf{G}_0\mathbf{f}_0 + \mathbf{G}_0^2\mathbf{y}_0\right)t^2 \\
&\quad + \frac{1}{6}\left(\mathbf{f}_0'' + \mathbf{G}_0\mathbf{y}_0'' + 2\mathbf{G}_0'\mathbf{y}_0' + \mathbf{G}_0''\mathbf{y}_0 + 2\mathbf{G}_0\mathbf{f}_0' + 2\mathbf{G}_0^2\mathbf{y}_0' + 2\mathbf{G}_0\mathbf{G}_0'\mathbf{y}_0\right. \\
&\qquad \left. + \mathbf{G}_0^2\mathbf{f}_0 + \mathbf{G}_0^3\mathbf{y}_0 + \mathbf{G}_0'\mathbf{f}_0 + \mathbf{G}_0'\mathbf{G}_0\mathbf{y}_0\right)t^3 + \mathcal{O}(t^4) \\
&\equiv \mathbf{C}t + \mathbf{D}t^2 + \mathbf{E}t^3 + \mathcal{O}(t^4),
\end{aligned}
\tag{50}
$$

where we have introduced $\mathbf{C}, \mathbf{D}$, and $\mathbf{E}$ as the coefficients of $t, t^2$, and $t^3$ respectively. Then,

$$
\begin{aligned}
e^{-\mathbf{M}(t)}\int_0^t e^{\mathbf{M}(\tau)}\mathbf{z}(\tau)d\tau &= \left(\mathbf{I} - \mathbf{G}_0 t + \mathbf{A}_- t^2 + \mathcal{O}(t^3)\right)\left(\mathbf{C}t + \mathbf{D}t^2 + \mathbf{E}t^3 + \mathcal{O}(t^4)\right) \\
&= \left(\mathbf{f}_0 + \mathbf{G}_0\mathbf{y}_0\right)t + \frac{1}{2}\left(\mathbf{f}_0' + \mathbf{G}_0\mathbf{y}_0' + \mathbf{G}_0'\mathbf{y}_0 - \mathbf{G}_0\mathbf{f}_0 - \mathbf{G}_0^2\mathbf{y}_0\right)t^2 \\
&\quad + \frac{1}{6}\left(\mathbf{f}_0'' + \mathbf{G}_0\mathbf{y}_0'' + 2\mathbf{G}_0'\mathbf{y}_0' + \mathbf{G}_0''\mathbf{y}_0 - \mathbf{G}_0\mathbf{f}_0' - \mathbf{G}_0^2\mathbf{y}_0'\right. \\
&\qquad \left. - \mathbf{G}_0\mathbf{G}_0'\mathbf{y}_0 + \mathbf{G}_0^2\mathbf{f}_0 + \mathbf{G}_0^3\mathbf{y}_0 - 2\mathbf{G}_0'\mathbf{f}_0 - 2\mathbf{G}_0'\mathbf{G}_0\mathbf{y}_0\right)t^3 + \mathcal{O}(t^4) \\
&\equiv \mathbf{F}t + \mathbf{H}t^2 + \mathbf{J}t^3 + \mathcal{O}(t^4),
\end{aligned}
\tag{51}
$$

where again we have introduced new variables for the coefficients: $\mathbf{F}, \mathbf{H}$, and $\mathbf{J}$. We define $\square_i = \square(t_i)$, and take $t_i = 0$ and $t = t_{i+1}$. This means that $\square_0 = \square_i$ and we can substitute $t$ with $\Delta_i$. Therefore, treating equation 43 as a fixed-point iteration problem gives:

$$
\begin{aligned}
\mathbf{y}^{(i+1)}(\Delta_i) &= e^{-\mathbf{M}(\Delta_i)}\left[\mathbf{y}(0) + \int_0^{\Delta_i} e^{\mathbf{M}(\tau)}\left(\mathbf{f}(\mathbf{y}^{(i)}(\tau), \mathbf{x}(\tau), \theta) + \mathbf{G}(\tau)\mathbf{y}^{(i)}(\tau)\right)d\tau\right] \\
&= \left(\mathbf{I} - \mathbf{G}_i\Delta_i + \mathbf{A}_-\Delta_i^2 - \mathbf{B}_-\Delta_i^3\right)\mathbf{y}_i + \mathbf{F}\Delta_i + \mathbf{H}\Delta_i^2 + \mathbf{J}\Delta_i^3 + \mathcal{O}(\Delta_i^4) \\
&= \mathbf{y}_i + \mathbf{f}_i\Delta_i + \frac{1}{2}\left(\mathbf{f}_i' + \mathbf{G}_i\mathbf{y}_i' - \mathbf{G}_i\mathbf{f}_i\right)\Delta_i^2 + \frac{1}{6}\left(\mathbf{f}_i'' + \mathbf{G}_i\mathbf{y}_i'' + 2\mathbf{G}_i'\mathbf{y}_i' - \mathbf{G}_i\mathbf{f}_i' - \mathbf{G}_i^2\mathbf{y}_i'\right. \\
&\quad \left. + \frac{1}{2}\mathbf{G}_i\mathbf{G}_i'\mathbf{y}_i + \mathbf{G}_i^2\mathbf{f}_i - 2\mathbf{G}_i'\mathbf{f}_i - \frac{1}{2}\mathbf{G}_i'\mathbf{G}_i\mathbf{y}_i\right)\Delta_i^3 + \mathcal{O}(\Delta_i^4).
\end{aligned}
\tag{52}
$$

As $\mathbf{z}(t) = \mathbf{f}(\mathbf{y}(t), \mathbf{x}(t), \theta) + \mathbf{G}(t)\mathbf{y}(t)$, equation 9 is:

$$
\begin{aligned}
\mathbf{y}_{i+1} &= e^{-\mathbf{G}_c\Delta_i}\mathbf{y}_i + \mathbf{G}_c^{-1}\left(\mathbf{I} - e^{-\mathbf{G}_c\Delta_i}\right)\left(\mathbf{f}_c + \mathbf{G}_c\mathbf{y}_c\right) \\
&= \left(\mathbf{I} - \mathbf{G}_c\Delta_i + \frac{1}{2}\mathbf{G}_c^2\Delta_i^2 - \frac{1}{6}\mathbf{G}_c^3\Delta_i^3\right)\mathbf{y}_i + \mathbf{G}_c^{-1}\left(\mathbf{G}_c\Delta_i - \frac{1}{2}\mathbf{G}_c^2\Delta_i^2 + \frac{1}{6}\mathbf{G}_c^3\Delta_i^3\right) \\
&\quad \times \left(\mathbf{f}_c + \mathbf{G}_c\mathbf{y}_c\right) + \mathcal{O}(\Delta_i^4) \\
&= \mathbf{y}_i + \left(\mathbf{f}_c + \mathbf{G}_c\left(\mathbf{y}_c - \mathbf{y}_i\right)\right)\Delta_i - \frac{1}{2}\left(\mathbf{G}_c\mathbf{f}_c + \mathbf{G}_c^2\left(\mathbf{y}_c - \mathbf{y}_i\right)\right)\Delta_i^2 + \frac{1}{6}\left(\mathbf{G}_c^2\mathbf{f}_c + \mathbf{G}_c^3\left(\mathbf{y}_c - \mathbf{y}_i\right)\right) \\
&\quad \Delta_i^3 + \mathcal{O}(\Delta_i^4),
\end{aligned}
\tag{53}
$$

with $\mathbf{f}_c = \frac{1}{2}\left(\mathbf{f}_i + \mathbf{f}_{i+1}\right), \mathbf{G}_c = \frac{1}{2}\left(\mathbf{G}_i + \mathbf{G}_{i+1}\right)$, and $\mathbf{y}_c = \frac{1}{2}\left(\mathbf{y}_i + \mathbf{y}_{i+1}\right)$ for $t_i \le t < t_{i+1}$.

The Taylor expansions of the midpoint approximations are:

$$\mathbf{f}_c = \mathbf{f}_i + \frac{1}{2}\mathbf{f}_i'\Delta_i + \frac{1}{4}\mathbf{f}_i''\Delta_i^2 + \mathcal{O}(\Delta_i^3)$$

$$\mathbf{G}_c = \mathbf{G}_i + \frac{1}{2}\mathbf{G}_i'\Delta_i + \frac{1}{4}\mathbf{G}_i''\Delta_i^2 + \mathcal{O}(\Delta_i^3) \tag{54}$$

$$\mathbf{y}_c = \mathbf{y}_i + \frac{1}{2}\mathbf{y}_i'\Delta_i + \frac{1}{4}\mathbf{y}_i''\Delta_i^2 + \mathcal{O}(\Delta_i^3).$$

Recall that $\mathbf{y}^{(i+1)}(\Delta_i)$ is the $(i+1)$th iterated guess of the form of $\mathbf{y}$, given $\mathbf{y}(0)$ and the previous guess $\mathbf{y}^{(i)}(t)$, evaluated at $t = t_{i+1} = \Delta_i$. Meanwhile, $\mathbf{y}_{i+1}$ is the value of the approximation to $\mathbf{y}^{(i+1)}$ at $t = t_{i+1} = \Delta_i$, given $\mathbf{y}(0)$ and the previous guess $\mathbf{y}^{(i)}(t)$. This gives the local truncation error as:

$$\text{LTE} = \mathbf{y}^{(i+1)}(\Delta_i) - \mathbf{y}_{i+1}$$

$$= [\mathbf{f}_i - \mathbf{f}_c - \mathbf{G}_c(\mathbf{y}_c - \mathbf{y}_i)]\Delta_i + \frac{1}{2}\left[\mathbf{f}_i' + \mathbf{G}_i\mathbf{y}_i' - \mathbf{G}_i\mathbf{f}_i + \mathbf{G}_c\mathbf{f}_c + \mathbf{G}_c^2(\mathbf{y}_c - \mathbf{y}_i)\right]\Delta_i^2$$

$$+ \frac{1}{6}\left[\mathbf{f}_i'' + \mathbf{G}_i\mathbf{y}_i'' + 2\mathbf{G}_i'\mathbf{y}_i' - \mathbf{G}_i\mathbf{f}_i' - \mathbf{G}_i^2\mathbf{y}_i' + \frac{1}{2}\mathbf{G}_i\mathbf{G}_i'\mathbf{y}_i + \mathbf{G}_i^2\mathbf{f}_i - 2\mathbf{G}_i'\mathbf{f}_i\right.$$

$$\left. - \frac{1}{2}\mathbf{G}_i'\mathbf{G}_i\mathbf{y}_i - \mathbf{G}_c^2\mathbf{f}_c - \mathbf{G}_c^3(\mathbf{y}_c - \mathbf{y}_i)\right]\Delta_i^3 + \mathcal{O}(\Delta_i^4)$$

$$= \left[\mathbf{f}_i - \mathbf{f}_i - \frac{1}{2}\mathbf{f}_i'\Delta_i - \frac{1}{4}\mathbf{f}_i''\Delta_i^2 - \frac{1}{2}\mathbf{G}_i\mathbf{y}_i'\Delta_i - \frac{1}{4}\mathbf{G}_i\mathbf{y}_i''\Delta_i^2 - \frac{1}{4}\mathbf{G}_i'\mathbf{y}_i'\Delta_i^2\right]\Delta_i$$

$$+ \frac{1}{2}\left[\mathbf{f}_i' + \mathbf{G}_i\mathbf{y}_i' - \mathbf{G}_i\mathbf{f}_i + \mathbf{G}_i\mathbf{f}_i + \frac{1}{2}\mathbf{G}_i\mathbf{f}_i'\Delta_i + \frac{1}{2}\mathbf{G}_i'\mathbf{f}_i\Delta_i + \frac{1}{2}\mathbf{G}_i^2\mathbf{y}_i'\Delta_i\right]\Delta_i^2$$

$$+ \frac{1}{6}\left[\mathbf{f}_i'' + \mathbf{G}_i\mathbf{y}_i'' + 2\mathbf{G}_i'\mathbf{y}_i' - \mathbf{G}_i\mathbf{f}_i' - \mathbf{G}_i^2\mathbf{y}_i' + \frac{1}{2}\mathbf{G}_i\mathbf{G}_i'\mathbf{y}_i + \mathbf{G}_i^2\mathbf{f}_i - 2\mathbf{G}_i'\mathbf{f}_i\right.$$

$$\left. - \frac{1}{2}\mathbf{G}_i'\mathbf{G}_i\mathbf{y}_i - \mathbf{G}_i^2\mathbf{f}_i\right]\Delta_i^3 + \mathcal{O}(\Delta_i^4)$$

$$= \frac{1}{12}\left[-\mathbf{f}_i'' - \mathbf{G}_i\mathbf{y}_i'' + \mathbf{G}_i'\mathbf{y}_i' + \mathbf{G}_i\mathbf{f}_i' + \mathbf{G}_i^2\mathbf{y}_i' + \mathbf{G}_i\mathbf{G}_i'\mathbf{y}_i - \mathbf{G}_i'\mathbf{f}_i - \mathbf{G}_i'\mathbf{G}_i\mathbf{y}_i\right]\Delta_i^3 + \mathcal{O}(\Delta_i^4). \tag{55}$$

Recalling from section 3 that $\mathbf{G}(t) = -\frac{\partial \mathbf{f}}{\partial \mathbf{y}}$, we define $\mathbf{f}_i^{(m,n)} = \left.\frac{\partial^{m+n}\mathbf{f}}{\partial \mathbf{y}^m \partial \mathbf{x}^n}\right|_{t=0}$ and expand the total derivatives:

$$\mathbf{f}_i' = \mathbf{f}_i^{(1,0)}\mathbf{y}_i' + \mathbf{f}_i^{(0,1)}\mathbf{x}_i'$$

$$\mathbf{f}_i'' = \mathbf{f}_i^{(1,0)}\mathbf{y}_i'' + \left(\mathbf{f}_i^{(2,0)}\mathbf{y}_i'\right)\mathbf{y}_i' + \left(\mathbf{f}_i^{(1,1)}\mathbf{y}_i'\right)\mathbf{x}_i' + \left(\mathbf{f}_i^{(1,1)}\mathbf{x}_i'\right)\mathbf{y}_i' + \left(\mathbf{f}_i^{(0,2)}\mathbf{x}_i'\right)\mathbf{x}_i' + \mathbf{f}_i^{(0,1)}\mathbf{x}_i''$$

$$\mathbf{G}_i = -\mathbf{f}_i^{(1,0)} \tag{56}$$

$$\mathbf{G}_i' = -\mathbf{f}_i^{(2,0)}\mathbf{y}_i' - \mathbf{f}_i^{(1,1)}\mathbf{x}_i'.$$

The local truncation error can therefore also be written:

$$\text{LTE} = \frac{1}{12}\left[-2\left(\mathbf{f}_i^{(2,0)}\mathbf{y}_i'\right)\mathbf{y}_i' - \left(\mathbf{f}_i^{(1,1)}\mathbf{y}_i'\right)\mathbf{x}_i' - 2\left(\mathbf{f}_i^{(1,1)}\mathbf{x}_i'\right)\mathbf{y}_i' - \left(\mathbf{f}_i^{(0,2)}\mathbf{x}_i'\right)\mathbf{x}_i' - \mathbf{f}_i^{(0,1)}\mathbf{x}_i''\right.$$

$$- \mathbf{f}_i^{(1,0)}\mathbf{f}_i^{(0,1)}\mathbf{x}_i' + \mathbf{f}_i^{(1,0)}\left(\mathbf{f}_i^{(2,0)}\mathbf{y}_i'\right)\mathbf{y}_i + \mathbf{f}_i^{(1,0)}\left(\mathbf{f}_i^{(1,1)}\mathbf{x}_i'\right)\mathbf{y}_i + \left(\mathbf{f}_i^{(2,0)}\mathbf{y}_i'\right)\mathbf{f}_i \tag{57}$$

$$\left. + \left(\mathbf{f}_i^{(1,1)}\mathbf{x}_i'\right)\mathbf{f}_i - \left(\mathbf{f}_i^{(2,0)}\mathbf{y}_i'\right)\mathbf{f}_i^{(1,0)}\mathbf{y}_i - \left(\mathbf{f}_i^{(1,1)}\mathbf{x}_i'\right)\mathbf{f}_i^{(1,0)}\mathbf{y}_i\right]\Delta_i^3 + \mathcal{O}(\Delta_i^4).$$

Accordingly, taking the midpoint values for the approximation enables us to obtain a cubic local truncation error, $\text{LTE} = K\Delta_i^3 + \mathcal{O}(\Delta_i^4)$.

| Interpolation | Error | Expressions |
|---|---|---|
| Midpoint | $O(\Delta^3)$ | $\mathbf{y}_{i+1} = \bar{\mathbf{G}}_i \mathbf{y}_i + \bar{z}_i$ with $\bar{\mathbf{G}}_i = \exp(-\mathbf{G}_i \Delta_i)$ and $\bar{z}_i = \mathbf{G}_i^{-1}(\mathbf{I} - \bar{\mathbf{G}}_i)\mathbf{z}_i$, where $\mathbf{G}_i = (\mathbf{G}(t_i) + \mathbf{G}(t_{i+1}))/2$ and $\mathbf{z}_i = (\mathbf{z}(t_i) + \mathbf{z}(t_{i+1}))/2$ |
| Left value | $O(\Delta^2)$ | $\mathbf{y}_{i+1} = \bar{\mathbf{G}}_i \mathbf{y}_i + \bar{z}_i$ with $\bar{\mathbf{G}}_i = \exp(-\mathbf{G}_i \Delta_i)$ and $\bar{z}_i = \mathbf{G}_i^{-1}(\mathbf{I} - \bar{\mathbf{G}}_i)\mathbf{z}_i$, where $\mathbf{G}_i = \mathbf{G}(t_i)$ and $\mathbf{z}_i = \mathbf{z}(t_i)$ |
| Right value | $O(\Delta^2)$ | $\mathbf{y}_{i+1} = \bar{\mathbf{G}}_i \mathbf{y}_i + \bar{z}_i$ with $\bar{\mathbf{G}}_i = \exp(-\mathbf{G}_i \Delta_i)$ and $\bar{z}_i = \mathbf{G}_i^{-1}(\mathbf{I} - \bar{\mathbf{G}}_i)\mathbf{z}_i$, where $\mathbf{G}_i = \mathbf{G}(t_{i+1})$ and $\mathbf{z}_i = \mathbf{z}(t_{i+1})$ |
| Linear | $O(\Delta^3)$ | $\mathbf{y}_{i+1} = e^{-\mathbf{M}(t)}\left[\mathbf{y}_i + \int_{t_i}^{t_{i+1}} e^{\mathbf{M}(\tau)}\mathbf{z}(\tau)\,d\tau\right]$ where $\mathbf{M}(t) = \mathbf{G}(t_i)(t - t_i) + \frac{\mathbf{G}(t_{i+1}) - \mathbf{G}(t_i)}{2(t_{i+1} - t_i)}(t - t_i)^2$ and $z(t) = z(t_i) + \frac{z(t_{i+1}) - z(t_i)}{t_{i+1} - t_i}(t - t_i)$ |
| Quadratic | $O(\Delta^5)$ | $\mathbf{y}_{i+1} = e^{-\mathbf{M}(t)}\left[\mathbf{y}_i + \int_{t_i}^{t_{i+1}} e^{\mathbf{M}(\tau)}\mathbf{z}(\tau)\,d\tau\right]$ where $\mathbf{M}(t) = \int_{t_i}^{t} \mathbf{G}(\tau)\,d\tau$ with $\mathbf{G}(\tau)$ and $z(\tau)$ are interpolated quadratically using the values at $t_{i-1}$, $t_i$, and $t_{i+1}$. |

Table 3: Error bounds and expressions to be evaluated in ODE for various types of interpolations.

### A.6 OTHER KINDS OF INTERPOLATIONS

The choice on midpoint interpolation was made based on the consideration of both error bound and the evaluation simplicity. Table 3 provide the error bounds and expression to be evaluated for various interpolations. From the table, it is clear that midpoint interpolation provides a good balance between error and ease-of-compute expression.

## B EXPERIMENTAL DETAILS

### B.1 CODE IN JAX

```
1  def deer_iteration(
2      invlin: Callable[[List[jnp.ndarray], jnp.ndarray, Any], jnp.ndarray],
3      func: Callable[[List[jnp.ndarray], Any, Any], jnp.ndarray],
4      shifter_func: Callable[[jnp.ndarray, Any], List[jnp.ndarray]],
5      p_num: int,
6      params: Any,  # gradable
7      xinput: Any,  # gradable
8      invlin_params: Any,  # gradable
9      shifter_func_params: Any,  # gradable
10     yinit_guess: jnp.ndarray,
11     max_iter: int = 100,
12     ) -> Tuple[jnp.ndarray, List[jnp.ndarray], Callable]:
13   # obtain the functions to compute the jacobians and the function
14   jacfunc = jax.vmap(jax.jacfwd(func, argnums=0), in_axes=(0, 0, None))
15   func2 = jax.vmap(func, in_axes=(0, 0, None))
16
17   dtype = yinit_guess.dtype
18   # set the tolerance
19   tol = 1e-7 if dtype == jnp.float64 else 1e-4
20
21   def iter_func(iter_inp):
22     err, yt, iiter = iter_inp
23     # yt: (nsamples, ndims)
```

```
24      ytparams = shifter_func(yt, shifter_func_params)
25      # [p_num] + (nsamples, ndims, ndims)
26      gts = [-gt for gt in jacfunc(ytparams, xinput, params)]  # FUNCEVAL
27      # rhs: (nsamples, ndims)
28      rhs = func2(ytparams, xinput, params)  # FUNCEVAL
29      rhs += sum([jnp.einsum("...ij,...j->...i", gt, ytp)
30                  for gt, ytp in zip(gts, ytparams)])  # GTMULT
31      # yt_next: (nsamples, ndims)
32      yt_next = invlin(gts, rhs, invlin_params)  # INVLIN
33      err = jnp.max(jnp.abs(yt_next - yt))  # checking convergence
34      return err, yt_next, iiter + 1
35
36  def cond_func(iter_inp) -> bool:
37      err, _, iiter = iter_inp
38      return jnp.logical_and(err > tol, iiter < max_iter)
39
40  err = jnp.array(1e10, dtype=dtype)  # very high initial error
41  gt = jnp.zeros((yinit_guess.shape[0], yinit_guess.shape[-1],
42                  yinit_guess.shape[-1]), dtype=dtype)
43  gts = [gt] * p_num
44  iiter = jnp.array(0, dtype=jnp.int32)
45  err, yt, iiter = jax.lax.while_loop(
46      cond_func, iter_func, (err, yinit_guess, iiter))
47  return yt
```

## B.2 HAMILTONIAN NEURAL NETWORK TRAINING WITH NEURALODE

Let's denote $\mathbf{s}$ as the states of the system. In the two-body case, the states are positions and velocities of the two masses, $\mathbf{s} = (x_1, y_1, v_{x1}, v_{y1}, x_2, y_2, v_{x2}, v_{y2})^T$. For typical works in learning physical systems, a neural network is designed to take the states $\mathbf{s}$ as the input and produces the states dynamics $\dot{\mathbf{s}}$ as the output. In those works, a set of states and the states dynamics are given as the training data, $\{..., (\mathbf{s}_i, \dot{\mathbf{s}}_i), ...\}$, so the neural network can just be trained without solving an ODE.

In our set up, the training data is only given as the states as a function of time $\{..., \mathbf{s}_i(t), ...\}$. To train the model with the given dataset, we need to roll out the states as a function of time $\mathbf{s}(t)$ using NeuralODE. In our case, the dataset is generated by generating the initial condition randomly and the dynamics are governed by the gravitational force. The initial conditions are chosen so the orbits of the two-body system does not diverge and the orbits are still close to a circle, to make the simulation numerically stable. The states are rolled out from $t = 0$ to $t = 10$ with 10,000 sampled time. Within the time period, the system makes about 2-4 rounds of orbit. There are 1000 rows of dataset generated and split into 800 for the training, 100 for the validation, and 100 for the test.

In order to speed up the training, we start from 20 time points at the beginning of the training and increase the number of time points by 20 every 50 training steps until it reaches 10k time points. We performed the training using ADAM optimizer (Kingma & Ba, 2014).

For the model, we are using Hamiltonian neural network (Greydanus et al., 2019) that consists of 6 linear layers with softplus activation except on the last linear layer. The hidden layers have 64 elements each. The input to the network has 8 elements (corresponding to the states $\mathbf{s}$) and the output of the network has only 1 element that corresponds to the Hamiltonian value.

For every training step during the training with DEER method, we save the predicted trajectory for every row of the dataset. The saved trajectory will be used as the initial guess of the DEER method for the next training step. The training was performed using ADAM optimizer with $10^{-3}$ learning rate. To speed up the training, we start from 20 time points at the beginning of the training and increase the number of time points by 20 every 1 epoch (50 training steps) until it reaches the maximum 10k time points. The loss function for the training is a simple mean squared error.

## B.3 TIME SERIES CLASSIFICATION WITH GRU

The dataset used in this case is Eigenworms (Brown et al., 2013). There are 259 worm samples with very long sequences (length $N = 17984$) in this dataset, which are then divided into train, validation

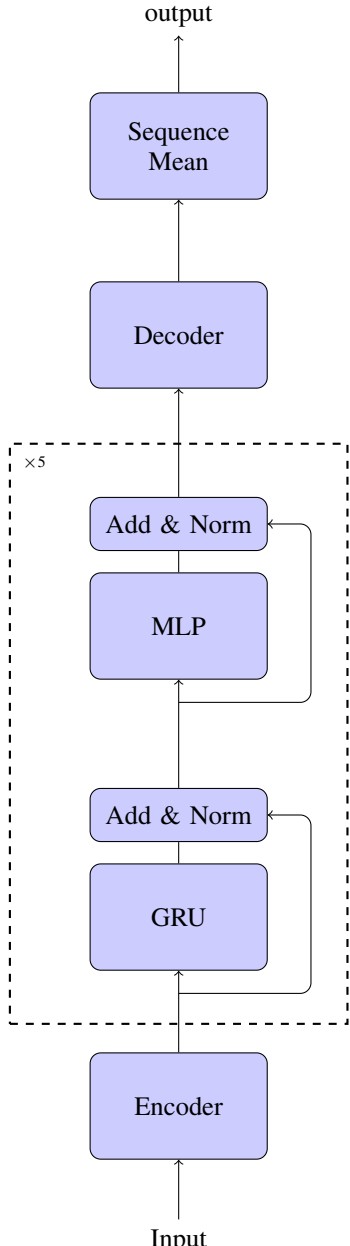

Figure 5: Architecture for the EigenWorms experiments.

and test sets using a 70%, 15%, 15% split following Morrill et al. (2021). Each sample is classified as either wild-type or one of four mutant types based on the representation using six base shapes.

The model used is shown in Figure 5. It consists of four main components: encoder, GRU, MLP and decoder. The encoder would project the input features to a higher dimension of 24, and the encoded input would then go through 5 layers of GRU and MLP pair, in that order. The number of hidden states for each GRU and MLP is set to 24. Then, the output of the GRU and MLP pair would go through a decoder to project it down to 5 classes. We then take the mean over the sequence length to obtain the final output. Residual connection followed by LayerNorm is applied to each GRU and MLP sublayer. Both the encoder and decoder are also simple MLPs, and all MLPs in the architecture have a depth of 1 with ReLU activation function in the hidden layer.

The training is done using cross entropy loss with the patience for early stopping set to 200 epochs per validation accuracy. The optimization algorithm we used is ADAM optimizer with $3 \times 10^{-4}$ learning rate and the gradient is clipped at 1.0 per global norm. The initial guesses for the DEER method are always initialized at zeros.

### B.4 SEQUENTIAL CLASSIFICATION WITH MULTI-HEAD GRU

The dataset we are using for this subsection is CIFAR-10 from torchvision. Each channel is normalized with mean $(0.4914, 0.4822, 0.4465)$ and standard deviation $(0.2023, 0.1994, 0.2010)$ and applying random horizontal flipping before transforming it to a sequence with 3 channels by simply reshaping the signal. Although the dataset from torchvision is already shuffled, we reshuffle the training dataset and split training and validation with 90% and 10% portions respectively. Our results that we present in this paper is the accuracy on the test dataset from torchvision using the checkpoint that gives the best validation accuracy.

For the architecture, we first use a linear layer to convert 3 channels into 256 channels, then we apply $M = 4$ composite layers of multi-head GRUs, and finally we apply another linear layer to convert from 256 channels into 10 channels for classification. Each composite layer of multi-head GRU consists of (1) a multi-head GRU with 32 heads, each with 8 channels with strides $\{2^0, 2^1, ..., 2^7\}$ uniformly assigned to each head, (2) a linear layer from 256 channels to 512 channels, (3) a gated linear unit (GLU) (Dauphin et al., 2017) to convert the channels back to 256, and (4) a skip connection followed by (5) a layer norm (Ba et al., 2016). Dropout with 0.1 drop probability was also inserted to the model. The total number of trainable parameters of the model is 1,347,082.

During the training, we use the cosine annealing with linear warmup for 100,000 training steps. The linear warmup stage took 10,000 steps to increase the learning rate from $10^{-7}$ to $2 \times 10^{-3}$ while the cosine annealing took the remaining 90,000 steps to get it down to $10^{-7}$. In each training step, we applied gradient clipping by global norm equals to 1.0. The training was performed using ADAMW optimizer (Loshchilov & Hutter, 2017) with 0.01 weight decay.

## C ADDITIONAL RESULTS

### C.1 EFFECT OF TOLERANCE LEVEL ON CONVERGENCE

The only hyperparameter of DEER is the tolerance level for the convergence criteria. In our implementation, if the maximum absolute deviation of the iterated values is below the tolerance level, then it is marked as "converge". The number of iterations to reach the convergence for float32 and float64 precision for various tolerance values are shown in Figure 6. From the figure, we can see that setting the tolerance level of $10^{-4}$ for float32 has the same number of convergence if setting the tolerance level of $3 \times 10^{-7}$. In both cases, either $10^{-4}$ or $3 \times 10^{-7}$ tolerance levels, the maximum absolute error of the DEER output with respect to the sequential method are both relatively low: $1.788 \times 10^{-7}$. This shows the insensitivity of the hyperparameter of DEER method to the convergence of the method.

### C.2 BENCHMARKS

Table 4 shows the speed up for batch size 16 and 8. It can be seen that the smaller batch size can achieve higher speed up on average.

Another interesting finding that we found can be seen in Figure 7. The figure shows the speed up comparison when using V100 GPU and when using A100 GPU. Although for smaller number of dimensions A100 can achieve larger speed up than V100, the speed up when using 32 dimensions on A100 suddenly drops to below 1. This effect is not as severe in V100.

Furthermore, we performed profiling to see where the bottleneck is. On the code in appendix B.1, we have denoted three lines of interest with FUNCEVAL, GTMULT, and INVLIN. Table 4 shows the run time using GRU with various number of dimensions with batch size of 16. From the table, we can see that the bottleneck is in solving $L_{\mathbf{G}}^{-1}$.

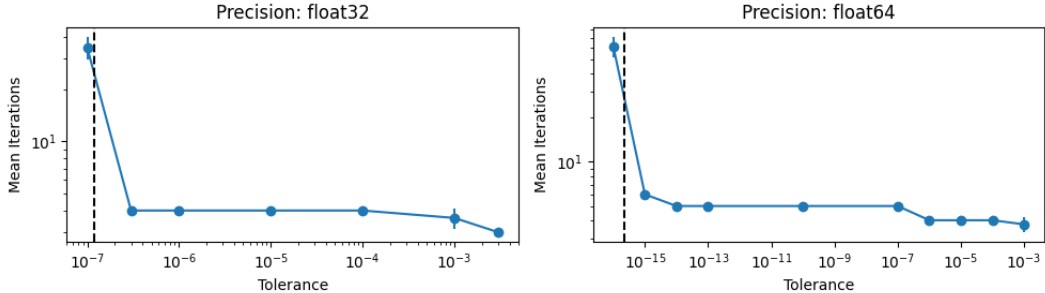

Figure 6: The mean number of iterations required to achieve the convergence for float32 and float64 precision. The case is an untrained GRU with 2 hidden size and 10000 sequence length. The mean and std of the plots were obtained from 16 data points each. The vertical black dashed lines show the 'eps' value of float32 and float64 ($1.1927 \times 10^{-7}$ and $2.22 \times 10^{-16}$ respectively).

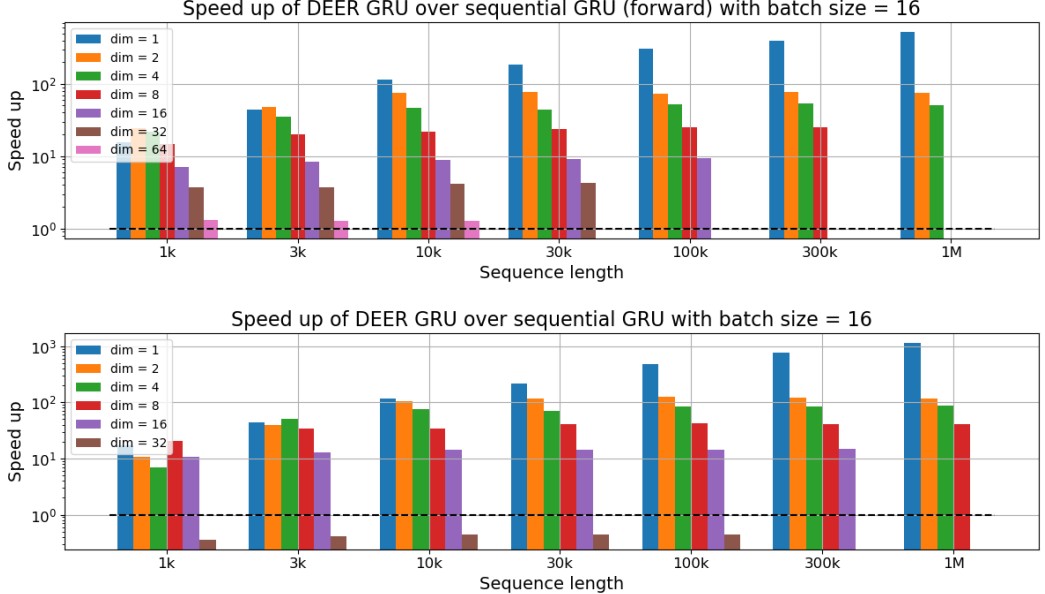

Figure 7: Speed up of DEER GRU over sequential method achieved in (top) V100 and (bottom) A100.

Table 4: The speed up of GRU calculated using DEER method (this paper) vs commonly-used sequential method on a V100 GPU with batch size of 16, 8, 4, and 2. The missing data for large number of dimensions and sequence lengths are due to insufficient memory in computing the DEER method. The numbers in the table represent the mean speed up over 5 different random seeds.

| #dims | Sequence lengths (batch size = 16) | | | | | | |
|---|---|---|---|---|---|---|---|
| | 1k | 3k | 10k | 30k | 100k | 300k | 1M |
| 1 | 15.7 | 43.5 | 114 | 182 | 309 | 394 | 516 |
| 2 | 24.4 | 47.6 | 75.0 | 78.4 | 72.4 | 77.1 | 74.6 |
| 4 | 22.1 | 35.4 | 46.5 | 44.4 | 52.1 | 54.2 | 51.3 |
| 8 | 14.8 | 20.0 | 21.9 | 24.0 | 25.2 | 25.1 | - |
| 16 | 7.11 | 8.37 | 8.89 | 9.23 | 9.46 | - | - |
| 32 | 3.74 | 3.76 | 4.16 | 4.24 | - | - | - |
| 64 | 1.29 | 1.26 | 1.27 | - | - | - | - |

| #dims | Sequence lengths (batch size = 8) | | | | | | |
|---|---|---|---|---|---|---|---|
| | 1k | 3k | 10k | 30k | 100k | 300k | 1M |
| 1 | 16.9 | 44.9 | 131 | 264 | 521 | 604 | 945 |
| 2 | 22.5 | 58.0 | 113 | 149 | 119 | 141 | 138 |
| 4 | 24.1 | 50.7 | 79.0 | 95.9 | 101 | 103 | 104 |
| 8 | 21.4 | 31.9 | 42.3 | 45.1 | 47.9 | 49.6 | 49.6 |
| 16 | 11.7 | 15.3 | 16.8 | 18.5 | 18.6 | 18.7 | - |
| 32 | 6.32 | 7.72 | 8.02 | 8.32 | 8.30 | - | - |
| 64 | 2.64 | 2.67 | 2.78 | 2.54 | - | - | - |

| #dims | Sequence lengths (batch size = 4) | | | | | | |
|---|---|---|---|---|---|---|---|
| | 1k | 3k | 10k | 30k | 100k | 300k | 1M |
| 1 | 16.6 | 45.1 | 130 | 316 | 809 | 1110 | 1530 |
| 2 | 23.1 | 62.7 | 162 | 256 | 301 | 294 | 306 |
| 4 | 24 | 63.1 | 131 | 174 | 171 | 199 | 207 |
| 8 | 23.1 | 48.1 | 73.9 | 89.3 | 95.7 | 98.4 | 100 |
| 16 | 18.2 | 28.6 | 32.7 | 35.1 | 37.4 | 39.1 | 38 |
| 32 | 11.8 | 14.9 | 15.9 | 17 | 17.3 | 15.2 | - |
| 64 | 5.14 | 5.7 | 5.62 | 5.22 | - | - | - |

| #dims | Sequence lengths (batch size = 2) | | | | | | |
|---|---|---|---|---|---|---|---|
| | 1k | 3k | 10k | 30k | 100k | 300k | 1M |
| 1 | 17.2 | 43.1 | 128 | 357 | 1030 | 1860 | 2660 |
| 2 | 24.8 | 66.7 | 187 | 382 | 589 | 516 | 610 |
| 4 | 25.2 | 64 | 172 | 280 | 372 | 416 | 433 |
| 8 | 24.4 | 60.5 | 114 | 162 | 178 | 181 | 202 |
| 16 | 22.5 | 45.5 | 62.8 | 68.5 | 75.3 | 78.1 | 77.7 |
| 32 | 16.7 | 25.7 | 32 | 31.4 | 34.3 | 35.1 | - |
| 64 | 9.18 | 10.4 | 11.1 | 10.4 | 10.7 | - | - |

Table 5: The run time of one iteration of a GRU cell in a V100 GPU in nanoseconds. Numbers that are less than 30 $\mu$s are not present in our profiling tool.

| Line label | Number of dimensions | | | | | |
|---|---|---|---|---|---|---|
| | 1 | 2 | 4 | 8 | 16 | 32 |
| FUNCEVAL | 31,430 | 432,184 | 911,824 | 1,194,010 | 3,419,370 | 5,249,474 |
| GTMULT | | 88,256 | 248,562 | 428,030 | 1,357,965 | 4,724,395 |
| INVLIN | 147,669 | 1,336,836 | 2,015,883 | 4,460,318 | 9,786,933 | 19,248,959 |

Table 6: GPU memory consumption in MiB for evaluating GRU using DEER method with batch size = 16 and for various number of dimensions.

| Number of dimensions | 1 | 2 | 4 | 8 | 16 | 32 |
|---|---|---|---|---|---|---|
| GPU memory (MiB) | 18.32 | 73.25 | 161.14 | 380.87 | 1351.68 | 5038.08 |

Figure 8: (a) The wall-clock comparison and (b) training step comparison of training LEM with DEER and with sequential method using the same amount of memory by increasing the batch size for sequential method.

We also did benchmark the GPU memory consumption as a function of number of dimensions with batch size equals to 16 in the GRU case. The results can be seen in Table 6. From the table, it can be seen that the GPU memory consumption approximately grows quadratically, as expected from storing the matrices $\mathbf{G}$ explicitly.

### C.3  DEER VS SEQUENTIAL WITH THE SAME MEMORY CONSUMPTION

To make another fair comparison, we ran an experiment comparing DEER vs sequential method using the same memory consumption. In this case, we use LEM architecture (Rusch et al., 2021) on EigenWorms (Brown et al., 2013) dataset. All of the hyperparameters are set to be the same, except the batch size. We increase the batch size for sequential method to match the memory consumption of DEER. With DEER, we set the batch size equals to 3 while for sequential method, we set it to 70. The GPU memory usage for both cases is about 2.6 GB GPU.

The results can be seen in 8. From the figure, we can see that although DEER uses smaller batch size and requires more steps to converge, it can achieve desirable results faster than using sequential method in terms of wall-clock time.

