# OpenReview forum: "Parallelizing non-linear sequential models over the sequence length"
_ICLR.cc/2024/Conference — ICLR 2024 poster_

### Official Review · Reviewer_gw76 · 2023-10-23

**Soundness:** 3 good
**Presentation:** 3 good
**Contribution:** 3 good
**Rating:** 6
**Confidence:** 3

**Summary:**

This paper proposes parallelizing evaluation and training of nonlinear sequential models using fixed point iteration methods. The paper proposes a method that restates nonlinear differential equations as fixed-point iterations with quadratic convergence, as in Newton's root finding method. The result is the DEER method. Speed and performance results are presented for a NeuralODE and GRU that are parallelized using DEER. Theoretical equivalences and convergence rates are established in the appendix.

**Strengths:**

- The proposed DEER method is well-motivated and presented clearly

- The theoretical results seem sound, though I only skimmed the proofs to follow the arguments, rather than check every detail line-by-line

- The method is general and can be applicable to a host of nonlinear differential equation methods such as neural ODEs and any nonlinear RNN (e.g. LSTM, GRU, etc) of broad interest to the sequence modeling community.

- The method has the advantage of theoretically having quadratic convergence. Experimental results support the claim that this method should lead to nearly equivalent results (up to numerical precision) as evaluating the nonlinear ODE sequentially

- For small hidden sizes, the empirical speed ups are substantial

**Weaknesses:**

- The biggest weakness of the paper is the empirical results, particularly related to performance. Only two tasks are considered, a synthetic physics system where HNNs are trained and the EigenWorms task where a GRU is trained.
  - For the EigenWorms task, the GRU does not significantly perform that much better than several of the baselines and is outperformed by others.  It is claimed that the DEER method enables faster training and thus experimentation to identify optimal GRU architectures. But is this the best the GRU could do?
  -  It would have been interesting to see the DEER method applied to some of the other methods such as LEM or UnICORNN since they could presumably be trained with the DEER method also. Would they maintain the same performance? Could better performance be achieved with the DEER method since they could potentially be trained faster and thus be hyperparameter tuned more? An exploration of this would strengthen the paper and broaden the impact.
  - A greater variety of tasks would increase the impact of the paper and potentially broaden the audience. E.g. there are many other common sequential "long sequence" tasks ranging from sequential MNIST, sequential CIFAR, to the LRA benchmarks. It would have been interesting to see how a nonlinear RNN such as GRU performed on LRA.

- There are many interesting theoretical results pushed to the Appendix. The authors might potentially consider packaging some of the formal statements as propositions in the main paper, to signpost the results and guide the reader to understand the rigor behind the method as well as point to the location of the proofs more explicitly in the Appendix.

- A weakness of the proposed method is the cubic complexity in the hidden size that the DEER method incurs. This limits scalability and the applicability of the method to larger scale systems. This is mentioned, but perhaps a further discussion of potential paths to address this could be interesting and helpful.

**Questions:**

1. Can you somehow quantify the claim that the DEER method can improve the performance of the GRU since it allows more hyperparameter tuning? Perhaps report the results of a single sequential GRU run, and then show the improvement that can be achieved by hyperparameter tuning the GRU for multiple runs (allowed by the parallelization of the DEER method).

2. Can you apply the DEER method to LEM and/or UniCORNN and report the results and compare the runtimes?

3. Can we add more tasks? Is it possible to run a GRU or other nonlinear RNN on Long Range Arena? Or does the cubic complexity of DEER prevent running large enough models for some of these tasks? Can we at least add other common RNN benchmarks such as sequential CIFAR?

4. Can you add a PDE example (discussed in the appendix)?

---

> ### Author Response · Authors · 2023-11-20
> **Response to Reviewer gw76**
>
> Thank you for your reviews and comments. Below is our responses to your questions.
>
> > Can you somehow quantify the claim that the DEER method can improve the performance of the GRU since it allows more hyperparameter tuning? Perhaps report the results of a single sequential GRU run, and then show the improvement that can be achieved by hyperparameter tuning the GRU for multiple runs (allowed by the parallelization of the DEER method).
>
> We ran hyperparameters tuning during the discussion period as the reviewer suggested and we can achieve the improvement for GRU from 82.1% to 88.0%, beating a more modern technique Neural RDE. We will update the results in the paper.
>
> [comment]: <> (<p style="color:blue">[TODO: put in the paper ???]</p>)
>
> > Can you apply the DEER method to LEM and/or UniCORNN and report the results and compare the runtimes?
>
> We have ran experiments using DEER on LEM. It is more challenging to reproduce their results than we thought as running the code from the official repository does not really reproduce the results on the paper. So we used DEER method to tune the hyperparameters efficiently and we can obtain similar results to their paper. With DEER, one experiment can run in less than 30 minutes, while with the code from the official repository, it took us 6 hours to run an experiment.  We will put the results in the paper.
>
> [comment]: <> (We have added the results in our paper in [???])
>
> [comment]: <> (<p style="color:blue">[TODO: write the results for LEM in our paper ???]</p>)
>
> > Can we add more tasks? Is it possible to run a GRU or other nonlinear RNN on Long Range Arena? Or does the cubic complexity of DEER prevent running large enough models for some of these tasks? Can we at least add other common RNN benchmarks such as sequential CIFAR?
>
> Yes, we have added a new experiment result on sequential CIFAR-10 (sCIFAR-10) using multiple head GRU with different strides. We can get around the cubic complexity of DEER by splitting the GRU into multiple heads. For example, instead of using correlated 256 states, we split them into 16 heads with 16 states each. We ran an experiment with this idea on sequential CIFAR-10 where we elaborate in the "Response to all reviewers" section 1 above. We will put the results in the paper.
>
> > Can you add a PDE example (discussed in the appendix)?
>
> Although DEER can be used for PDE, the focus of our paper is for one dimensional differential equation (i.e. ODE or discrete difference equation). Applying it for a PDE would require writing the solver of the linear equation which would take a significant amount of effort. Exploring this direction is a possible future work. For now, we put an example on how to apply DEER framework to a simple PDE case in the Appendix.

---

> > ### Comment · Reviewer_gw76 · 2023-11-21
> >
> > Thank you for the response and additional experiments. I have increased my score.

---

> > > ### Author Response · Authors · 2023-11-21
> > > **Thank you**
> > >
> > > Thank you for your response and for the score increase!

---

### Official Review · Reviewer_EMpw · 2023-11-01

**Soundness:** 4 excellent
**Presentation:** 3 good
**Contribution:** 2 fair
**Rating:** 6
**Confidence:** 4

**Summary:**

This paper proposes a novel parallelization scheme for sequential models. The method works by recasting the model as a fixed point problem and using Newton's method to solve it in parallel. Involves computing the Jacobian and Hessian of the model explicitly to enable quadratic convergence. An adjoint method is developed to compute the gradient in parallel. Experimental results are shown in speeding up RNNs and neural ODEs, showing significant speedups especially for long sequence lengths. Extensive proofs are given.

**Strengths:**

+ The method clearly has large speedups in certain training regimes, particularly for long sequences and small batch sizes
+ The theory of the method is clearly described, with proofs provided in the appendix.
+ The proofs in the appendix are clearly described.
+ The method is generally applicable to any sequential method, unlike previous methods which require specific architectures or structural assumptions.
+ Convergence is quicker than previous methods which did not incorporate Jacobians
+ Potentially, the method could also be applied at inference time to speed up sequence generation, not just at training time.

**Weaknesses:**

+ The practical importance of this method is somewhat unclear.
  From a practical point of view, the DEER method is a mechanism to use more memory in order to speed up the forward and backwards pass.
  Therefore, many of the experiments, particularly figure 2, are not really a fair comparison, as it's very common to increase the batch size until the memory is fully utilized.
  In other words, the fairer comparison would be to fix the throughput (i.e. FLOP/s or memory usage) of DEER and the sequential method the same and to compare the two methods. I think it's quite plausible that for a given throughput, using DEER to do more minibatch steps
  with smaller batch size can result in faster convergence. However, I don't believe this argument is made in the paper. Figure 4 gets close, but I don't think this is normalized in terms of
  throughput.
+ The increased memory usage of DEER at higher hidden dimensions seems a big stumbling block. As far as I am aware, a hidden dimension of 8 would be considered very small and impose a limit on the expressiveness
  of a recurrent model. This issue is not really addressed in the work.

**Questions:**

+ Can you provide comparisons to the sequential method when the throughput of the GPU is held constant (e.g. the memory is approximately fully utilized)?
+ How do RNNs with smaller hidden sizes compare to those with larger hidden sizes? Is it the case that the increased speed of DEER is able to offset for a reduction in hidden size that might be needed to use DEER?
+ Would you be able to use an approximation for the Jacobian (low-rank, etc) to reduce the memory usage?
+ How many iterations are typically required to converge? How does this compare against zero-order methods that don't use a Jacobian?

---

> ### Author Response · Authors · 2023-11-20
> **Response to Reviewer EMpw**
>
> Thank you for your reviews and comments. Below is our responses to your comments and questions.
>
> > Can you provide comparisons to the sequential method when the throughput of the GPU is held constant (e.g. the memory is approximately fully utilized)?
>
> We ran an experiment using LEM on EigenWorms to compare DEER and sequential methods. To make the memory consumption similar, we increase the batch size when running with the sequential method. As the result, although DEER method took more steps to achieve the similar results as sequential method, it was 3 times faster in terms of wall-clock time. This confirms Reviewer EMpw's hypothesis that *"for a given throughput, using DEER to do more minibatch steps with smaller batch size can result in faster convergence"*. We will put the results in the paper.
>
> [comment]: <> (<p style="color:blue">[??? TODO: put in the paper]</p>)
>
> > How do RNNs with smaller hidden sizes compare to those with larger hidden sizes? Is it the case that the increased speed of DEER is able to offset for a reduction in hidden size that might be needed to use DEER?
>
> The unfavourable scaling of DEER with respect to the hidden size can be avoided by having multiple “heads” where each head has a smaller hidden size. We ran a new experiment with this idea on sequential CIFAR-10 where we elaborate in the "Response to all reviewers" section 1 above.
>
> > Would you be able to use an approximation for the Jacobian (low-rank, etc) to reduce the memory usage?
>
> Yes! In fact, this is the area that we would like to explore as a continuation of this work.
>
> > How many iterations are typically required to converge? How does this compare against zero-order methods that don't use a Jacobian?
>
> With a typical untrained GRU with 16 hidden sizes, 10000 sequence length, float32, it usually takes about 4 iterations to converge (i.e. max absolute error < 1e-4). With zero-order methods, it typically takes about 23-26 for an untrained GRU. Despite it not looking very different for untrained GRU, the number of zero-order iteration could significantly increase with different settings. For example, multiplying all the parameters of untrained GRU by 5 would get the numbers to about 7 for our method and 330 - 360 for the zero order method.

---

> > ### Comment · Reviewer_EMpw · 2023-11-22
> > **Response to rebuttal**
> >
> > Thank you for your response. I'm glad to see that the fairer comparisons still resulted in favourable results for DEER.
> > The idea of using multiple heads for the RNNs is an interesting one, and a nice approach to sidestep the unfavourable scaling of the DEER method with hidden size.
> >
> > Based on the rebuttal and responses, I'm increasing my score.

---

> ### Author Response · Authors · 2023-11-22
>
> We kindly request your attention to our recent comments and updates, and would greatly appreciate your response before the end of the discussion period on Thursday, 23 November 2023 at 11:59 GMT. Thank you for your time and consideration.

---

### Official Review · Reviewer_QySF · 2023-11-03

**Soundness:** 3 good
**Presentation:** 2 fair
**Contribution:** 3 good
**Rating:** 6
**Confidence:** 4

**Summary:**

This paper introduces a method for parallel evaluation of non-linear sequence models (neural differential equations, GRUs) as a limit case of differentiable multiple shooting. The key idea is to linearize the multiple shooting update equation and use exact parallel methods (e.g., parallel scans) to evaluate each step of the root-finding procedure. The method is evaluated on time series classifiction, hamiltonian neural ODE training, and benchmarked for efficiency.

**Strengths:**

* The method is clearly presented. The authors do a good job contextualizing the approach with respect to direct multiple shooting, including recent work on adapting it to Neural ODEs (Appendix A.1, A.2)
* The efficiency evaluation is quite thorough, investigating the effect of batch size, state dimension and sequence length.

**Weaknesses:**

* It is not clear whether this approach improves over direct multiple shooting (which is also applicable to GRUs). Is there some drawback to the linearization you introduce? Do you require more steps to converge?
* The benchmarking is quite limited, both tasks showcased are small scale. The tasks do a good job at showing relative performance (end-to-end time) improvements, but they do not provide any insight on the method itself. Are there important hyperparameters, methods that could impact the final results and convergence? You mention mid-point in passing in Sec 3.3, could you elaborate on the choice of other interpolation methods that you have tried?

**Questions:**

* How is the recurrent step baseline implemented here? Since you use JAX, have you tried to otimize the recurrence (jit-compile, or fuse)?
* Although the scaling of the proposed method in with state dimension is quite unfavourable, many existing state-of-the-art sequence models (S4, H3, Hyena, RWKV) take a specific "multi-SISO" (single-input, single-output) form, where each channel of the model (in width) is assigned its own state. Although these models typically use linear recurrences that can simply apply the convolution theorem or a parallel-scan, the same structure can be used in the nonlinear case. Since these models are usually trained with small state dimensions (8 - 64), this would correspond to integrating many more systems in parallel (model width times batch size, rather than batch size as in the GRU case). Can the authors commend on whether DEER would be applicable in this case (and ideally show some preliminary result on a simple task?)
* How many steps do you need to converge with DEER? How much higher is the latency of one DEER evaluation vs one step of a corresponding GRU?
* Can you comment on the different rates of approximation of this linearized version of multiple shooting, and regular multiple shooting?

---

> ### Author Response · Authors · 2023-11-20
> **Response to Reviewer QySF**
>
> Thank you for your review on our paper. Here are our response to your comments and questions.
>
> > It is not clear whether this approach improves over direct multiple shooting (which is also applicable to GRUs). Is there some drawback to the linearization you introduce? Do you require more steps to converge?
>
> In regards to the linearization we refer to in the appendix, we realise that we made a mistake: the direct multiple shooting layer (MSL) equation (eq. 17 in our paper) is actually already a linear equation ($b_{i+1}^{(k+1)}$ is linear w.r.t. $b_i^{(k+1)}$), so we didn’t actually do linearization over MSL because it’s already linear. We apologise for the mistake. As they are both linear and can use Newton’s method, the number of iterations to converge should be the same.
>
> To show the advantage of our method over the multiple shooting layer (MSL), we ran the runtime comparisons between DEER, multiple shooting layers from torchdyn (maintained by the authors of the MSL paper: S. Massaroli, *et al.*, 2021), and our JAX implementation of MSL from torchdyn. In the comparison, we used a simple ODE case with 10,000 time points.
>
> | Method | Device | Runtime | Notes |
> |-|-|-|-|
> | **DEER** | **GPU** | **1.62 ms** | |
> | torchdyn's MSL | GPU | 12.47 s | 20 sections |
> | torchdyn's MSL | GPU | 11.80 s | 200 sections |
> | JAX MSL | GPU | 202 ms | 20 sections |
> | JAX MSL | GPU | 117 ms | 200 sections |
> | DEER | CPU | 105 ms | |
> | torchdyn's MSL | CPU | 4.59 s | 20 sections |
> | torchdyn's MSL | CPU | 3.60 s | 200 sections |
> | JAX MSL | CPU | 8.39 ms | 20 sections |
> | **JAX MSL** | **CPU** | **7.32 ms** | **200 sections** |
>
> Although our method is slower in CPU, DEER can take more parallelization advantage with GPU. We will put this result in the paper.
>
> > Are there important hyperparameters, methods that could impact the final results and convergence?
>
> DEER has only one hyperparameter to tune: the tolerance for convergence. However, as the method has quadratic convergence rate, the effect of the tolerance value is not significant to the number of iterations to reach the convergence.
>
> > You mention mid-point in passing in Sec 3.3, could you elaborate on the choice of other interpolation methods that you have tried?
>
> Yes, we explored other kinds of interpolation methods, including: (1) left value, (2) right value, (3) linear interpolation, and (4) quadratic interpolation with Simpson’s rule. Interpolation (1) and (2) gives a quadratic error ($O(\Delta t^2)$). Interpolation (3) gives cubic error ($O(\Delta t^3)$), but the analytical expression of equation (8) becomes complicated. Interpolation (4) gives a fifth-power error ($O(\Delta t^5)$), but it gives a very complicated analytical expression of equation (8). We end up with midpoint interpolation because it gives reasonably low error (cubic error) with a very simple expression of equation (8). We will put the details on the interpolations in the paper.
>
> > How is the recurrent step baseline implemented here?
>
> Yes, we used jax.lax.scan and jit-compile (which I believe includes operator fusion) for the sequential method.
>
> > Although the scaling of the proposed method in with state dimension is quite unfavourable, many existing state-of-the-art sequence models (S4, H3, Hyena, RWKV) take a specific "multi-SISO" (single-input, single-output) form, where each channel of the model (in width) is assigned its own state. ... Can the authors commend on whether DEER would be applicable in this case (and ideally show some preliminary result on a simple task?)
>
> We agree with the reviewer’s suggestion on having multiple independent states. We can group the channel dimension into several “heads”, thus avoiding the cubic scaling for DEER. We ran an experiment with this idea on sequential CIFAR-10 where we elaborate in the "Response to all reviewers" section 1 above.
>
> > How many steps do you need to converge with DEER? How much higher is the latency of one DEER evaluation vs one step of a corresponding GRU?
>
> With a typical untrained GRU, it usually takes about 4 iterations, but during the training it can take more steps to converge. For a GRU of input and hidden sizes of 16, batch size 16, sequence length 10K, a DEER iteration typically takes 5.4 ms, while one step of GRU (from one sequence point to another sequence point) takes 22 μs. Although one DEER iteration takes much longer than one sequential step, the number of required iterations is usually much less than the number of sequence steps.
>
> > Can you comment on the different rates of approximation of this linearized version of multiple shooting, and regular multiple shooting?
>
> This is related to our earlier comment regarding linearization. We made a mistake by saying that we do linearization over direct multiple shooting Newton’s method update step, while the direct multiple shooting update step is actually already linear.

---

> ### Author Response · Authors · 2023-11-22
>
> We kindly request your attention to our recent comments and updates, and would greatly appreciate your response before the end of the discussion period on Thursday, 23 November 2023 at 11:59 GMT. Thank you for your time and consideration.

---

### Author Response · Authors · 2023-11-20
**Response to all reviewers: new experiments and results**

We thank all the reviewers for their comments and questions. There is a typical main concern from the 3 reviewers which is about lacking experimental variations. Therefore, we would like to use this space to inform the reviewers about the new experiments and new results that we obtained during the discussion period.

Please note that no changes in the paper has been made yet. We need more time to add new results and arrange the content to fit within the 9-page limit.

### 1. Sequential CIFAR-10 with multihead GRU

To address the concerns about the cubic scaling of DEER method, we show that this can be avoided using multihead GRU. For example, instead of using a GRU with 256 channels, we divided them into a GRU with 32 heads with 8 channels each. The channels from different heads are detached, so the Jacobian matrix for each head is only $(8\times 8)$ instead of $(256\times 256)$. Moreover, each head can have different strides and in our case, we use strides from the set $\\{2^0, 2^1, ..., 2^7\\}$ (so there are 4 heads that have the same stride value). The idea of detaching channels is not new, it has been shown to work really well in state-space models like S4, S4D, Hyena, *etc*.

Our experiments with GRU + DEER can run about 3 - 5 times faster than with GRU + sequential method, allowing us to run more experiments during this discussion period. The best test set accuracy we obtained was 77.63%. Although this is not the state-of-the-art for sequential CIFAR-10, this is the best accuracy obtained by non-linear recurrent neural network (as shown in [PapersWithCode](https://paperswithcode.com/sota/sequential-image-classification-on-sequential-1)). **Please note that** the aim of our paper is not to introduce a new architecture, but to focus on a computational method that allows faster training for non-linear sequential models. We will put the results in the paper.

[comment]: <> (<p style="color:blue">[TODO: put in the paper ???]</p>)

### 2. EigenWorms with LEM

The state-of-the-art for EigenWorms dataset is currently held by LEM (T. Konstantin Rusch, *et al.*, 2022). Therefore, we ran experiments using DEER method with LEM architecture, trying to replicate their results. The replication attempt is not as easy as we expected because running the official code didn't give us the results presented in the paper, so we need to run with various sets of hyperparameters.

Running the official code (written in Pytorch, using sequential method) took us about 6 hours to complete 1 experiment, while using DEER in JAX only took us less than 30 minutes. This allows us to find the hyperparameters that can achieve the similar results in a short amount of time. We will put the results in the paper.

[comment]: <> (<p style="color:blue">[TODO: put in the paper ???]</p>)

### 3. Improving GRU on EigenWorms

We use this discussion period to find the hyperparameters that can improve the performance of GRU in EigenWorms. With the new hyperparameters, we can improve the results from 82.1% to 88.0%. Although it is not the state-of-the-art, we show that classical architecture such as GRU can beat Neural RDE, a more modern technique on learning with long sequence. We will update the results in the paper.

[comment]: <> (<p style="color:blue">[TODO: put in the paper ???]</p>)

---

### Author Response · Authors · 2023-11-21
**Updates on the paper**

We have updated our paper to follow the recommendations and our responses to the reviewers. Here are the list of changes we made in the paper:

### Main content

1. Added a new section on sequential CIFAR-10 experiment using multi-head GRU (section 4.4). Note that we obtained a new result that has higher accuracy than our previous response.
2. Updated Table 1 to reflect the latest results of our hyperparameter tuning of GRU and our reproducibility attempt at LEM.
3. Added a new paragraph (3rd paragraph in section 3.5) to elaborate the only 1 hyperparameter of DEER method.
4. Added a new paragraph (last paragraph in section 4.3) to briefly explain our reproducibility attempt at LEM for EigenWorms.

### Appendix

1. Added a new section on "Other kinds of interpolations" to list the other interpolations we considered in NeuralODE (section A.6)
2. Added a new section on experimental details of multi-head GRU on sequential CIFAR-10 (section B.4)
3. Added a new section on the effect of tolerance level on convergence, to explain the only 1 hyperparameter of DEER (section C.1)
4. Added a new section to present the results of DEER vs sequential method using the same amount of memory (section C.3)

---

### Meta-Review · Area_Chair_YJF8 · 2023-12-10

**Metareview:**

The paper presents a novel parallel algorithm for evaluating and training sequential models such as RNNs and Neural ODEs. Their method, based on parallelizing GPU evaluation, reportedly accelerates training without sacrificing accuracy. The approach does not require specific architecture changes, making it broadly applicable.

Strengths:
- The method's independence from model architecture increases its potential impact across various applications.
- The reported acceleration in training time is substantial, addressing a critical bottleneck in sequence modeling.
- The authors added new results in response to reviewer concerns, demonstrating the method's effectiveness.

Weaknesses:
- The experiments mainly focus on small-scale datasets, raising questions about scalability.
- The paper lacks a thorough comparison with existing methods like direct multiple shooting.
- The method's cubic complexity with respect to hidden size could limit its scalability for larger models. However, the authors convinced the reviewers that this could be potentially addressed.

During the rebuttal period, most reviewers increased their scores. AC votes with the acceptance of the paper.

**Justification For Why Not Higher Score:**

Limited experimental validation.

**Justification For Why Not Lower Score:**

The paper is sound and practically useful.

---

### Decision · Program_Chairs · 2024-01-16

Accept (poster)